# Inverse design of 3d molecular structures with conditional generative neural networks

Niklas W. A. Gebauer [1,2,3✉], Michael Gastegger [1,3], Stefaan S. P. Hessmann [1,2],
Klaus-Robert Müller [1,2,4,5] & Kristof T. Schütt [1,2✉]

The rational design of molecules with desired properties is a long-standing challenge in chemistry. Generative neural networks have emerged as a powerful approach to sample novel molecules from a learned distribution. Here, we propose a conditional generative neural network for 3d molecular structures with specified chemical and structural properties. This approach is agnostic to chemical bonding and enables targeted sampling of novel molecules from conditional distributions, even in domains where reference calculations are sparse. We demonstrate the utility of our method for inverse design by generating molecules with specified motifs or composition, discovering particularly stable molecules, and jointly targeting multiple electronic properties beyond the training regime.

[1] Machine Learning Group, Technische Universität Berlin, 10587 Berlin, Germany. [2] Berlin Institute for the Foundations of Learning and Data, 10587 Berlin, Germany. [3] BASLEARN—TU Berlin/BASF Joint Lab for Machine Learning, Technische Universität Berlin, 10587 Berlin, Germany. [4] Department of Artificial Intelligence, Korea University, Anam-dong, Seongbuk-gu, Seoul 02841, Korea. [5] Max-Planck-Institut für Informatik, 66123 Saarbrücken, Germany. ✉email: n.gebauer@tu-berlin.de; kristof.schuett@tu-berlin.de

I dentifying chemical compounds with particular properties is a critical task in many applications, ranging from drug design[1–3] over catalysis[4] to energy materials[5–8]. As an exhaustive exploration of the vast chemical compound space is infeasible, progress in these areas can benefit substantially from inverse design methods. In recent years, machine learning (ML) has been used to accelerate the exploration of chemical compound space[9–15]. A plethora of methods accurately predicts chemical properties and potential energy surfaces of 3d structures at low computational cost[16–27]. Here, the number of reference calculations required for training ML models depends on the size of the domain to be explored. Thus, naive exploration schemes may still require a prohibitive number of electronic structure calculations. Instead, chemical space has to be navigated in a guided way with fast and accurate methods to distill promising molecules.

This gives rise to the idea of inverse molecular design[28], where the structure-property relationship is reversed. Here, the challenge is to directly construct molecular structures corresponding to a given set of properties. Generative ML models have recently gained traction as a powerful, data-driven approach to inverse design as they enable sampling from a learned distribution of molecular configurations[29]. By appropriately restricting the distributions, they allow obtaining sets of candidate structures with desirable characteristics for further evaluation. These methods typically represent molecules as graphs or SMILES strings[30,31], which lack information about the three-dimensional structure of a molecule. Therefore, the same molecular graph can represent various spatial conformations that differ in their respective properties, e.g., due to intramolecular interactions (hydrogen bonds, long-range interactions) or different orientations of structural motifs (rotamers, stereoisomers). Beyond that, connectivity-based representations are problematic in chemical systems where bonding is ambiguous, e.g., in transition metal complexes, conjugated systems or metals. Relying on these abstract representations is ultimately a limiting factor when exploring chemical space.

Recently, generative models that enable sampling of 3d molecular configurations have been proposed. This includes specifically designed approaches to translate given molecular graphs to 3d conformations[32–38], map from coarse-grained to fine-grained structures[39], sample unbiased equilibrium configurations of a given system[40,41], or focus on protein folding[42–46]. In contrast, other models aim at sampling directly from distributions of 3d molecules with arbitrary composition[47–56], making them suitable for general inverse design settings. These models need to be biased towards structures with properties of interest, e.g., using reinforcement learning[51,52,56], fine-tuning on a biased dataset[48], or other heuristics[54].

Some of us have previously proposed G-SchNet[48], an autoregressive deep neural network that generates diverse, small organic molecules by placing atom after atom in Euclidean space. It has been applied in the 3D-Scaffold framework to build molecules around a functional group associated with properties of interest in order to discover novel drug candidates[54]. Such an approach requires prior knowledge about the relationship between functional groups and target properties and might prevent the model from unfolding its potential by limiting sampling to very specific molecules. G-SchNet has been biased by fine-tuning on a fraction of the training dataset containing all molecules with a small HOMO-LUMO gap[48]. For this, a sufficient amount of training examples in the target space is required. However, the most interesting regions for exploration are often those where reference calculations are sparse.

In this work, we propose conditional G-SchNet (cG-SchNet), a conditional generative neural network for the inverse design of molecules. Building on G-SchNet, the model learns conditional distributions depending on structural or chemical properties allowing us to sample corresponding 3d molecular structures. Our architecture is designed to generate molecules of arbitrary size and does not require the specification of a target composition. Consequently, it learns the relationship between the composition of molecules and their physical properties in order to sample candidates exhibiting given target properties, e.g., preferring smaller structures when targeting small polarizabilities. Previously proposed methods have been biased towards one particular set of target property values at a time by adjusting the training objective or data[48,51]. In contrast, our conditional approach permits searching for molecules with any desired set of target property values after training is completed. It is able to jointly target multiple properties without the need to retrain or otherwise indirectly constrain the sampling process. This provides the foundation for the model to leverage the full information of the training data resulting in increased generalization and data efficiency. We demonstrate that cG-SchNet enables the exploration of sparsely populated regions that are hardly accessible with unconditional models. To this end, we conduct extensive experiments with diverse conditioning targets including chemical properties, atomic compositions and molecular fingerprints. In this way, we generate novel molecules with predefined structural motifs, isomers of a given composition that exhibit specific chemical properties, and novel configurations that jointly optimize HOMO-LUMO gap and energy. This demonstrates that our model enables flexible, guided exploration of chemical compound space.

## Results

**Targeted 3d molecule generation with cG-SchNet.** We represent molecules as tuples of atom positions $\mathbf{R}_{\leq n} = (\mathbf{r}_1, \ldots, \mathbf{r}_n)$ with $\mathbf{r}_i \in \mathbb{R}^3$ and corresponding atom types $\mathbf{Z}_{\leq n} = (Z_1, \ldots, Z_n)$ with $Z_i \in \mathbb{N}$. cG-SchNet assembles these structures from sequences of atoms that are placed step by step in order to build the molecule in an autoregressive manner, where the placement of the next atom depends on the preceding atoms (Fig. 1a and c). In contrast to G-SchNet[48], which learns an unconditional distribution over molecules, cG-SchNet samples from target-dependent conditional probability distributions of 3d molecular structures (Fig. 1b).

Given a tuple of $k$ conditions $\mathbf{\Lambda} = (\lambda_1, \ldots, \lambda_k)$, cG-SchNet learns a factorization of the conditional distribution of molecules, i.e., the joint distribution of atom positions and atom types conditioned on the target properties:

$$p(\mathbf{R}_{\leq n}, \mathbf{Z}_{\leq n} | \mathbf{\Lambda}) = \prod_{i=1}^{n} p(\mathbf{r}_i, Z_i | \mathbf{R}_{\leq i-1}, \mathbf{Z}_{\leq i-1}, \mathbf{\Lambda}). \quad (1)$$

In fact, we can split up the joint probability of the next type and the next position into the probability of the next type and the probability of the next position given the associated next type:

$$p(\mathbf{r}_i, Z_i | \mathbf{R}_{\leq i-1}, \mathbf{Z}_{\leq i-1}, \mathbf{\Lambda})$$
$$= p(Z_i | \mathbf{R}_{\leq i-1}, \mathbf{Z}_{\leq i-1}, \mathbf{\Lambda}) \, p(\mathbf{r}_i | \mathbf{R}_{\leq i-1}, \mathbf{Z}_{\leq i}, \mathbf{\Lambda}). \quad (2)$$

This allows predicting the next type before the next position. We approximate the distribution over the absolute position from distributions over distances to already placed atoms

$$p(\mathbf{r}_i | \mathbf{R}_{\leq i-1}, \mathbf{Z}_{\leq i}, \mathbf{\Lambda}) = \frac{1}{\alpha} \prod_{j=1}^{i-1} p(r_{ij} | \mathbf{R}_{\leq i-1}, \mathbf{Z}_{\leq i}, \mathbf{\Lambda}) \quad (3)$$

which guarantees that it is equivariant with respect to translation and rotation of the input. Here $\alpha$ is the normalization constant and $r_{ij} = ||\mathbf{r}_i - \mathbf{r}_j||$ is the distance between the new atom $i$ and a previously placed atom $j$. This approximation has previously been

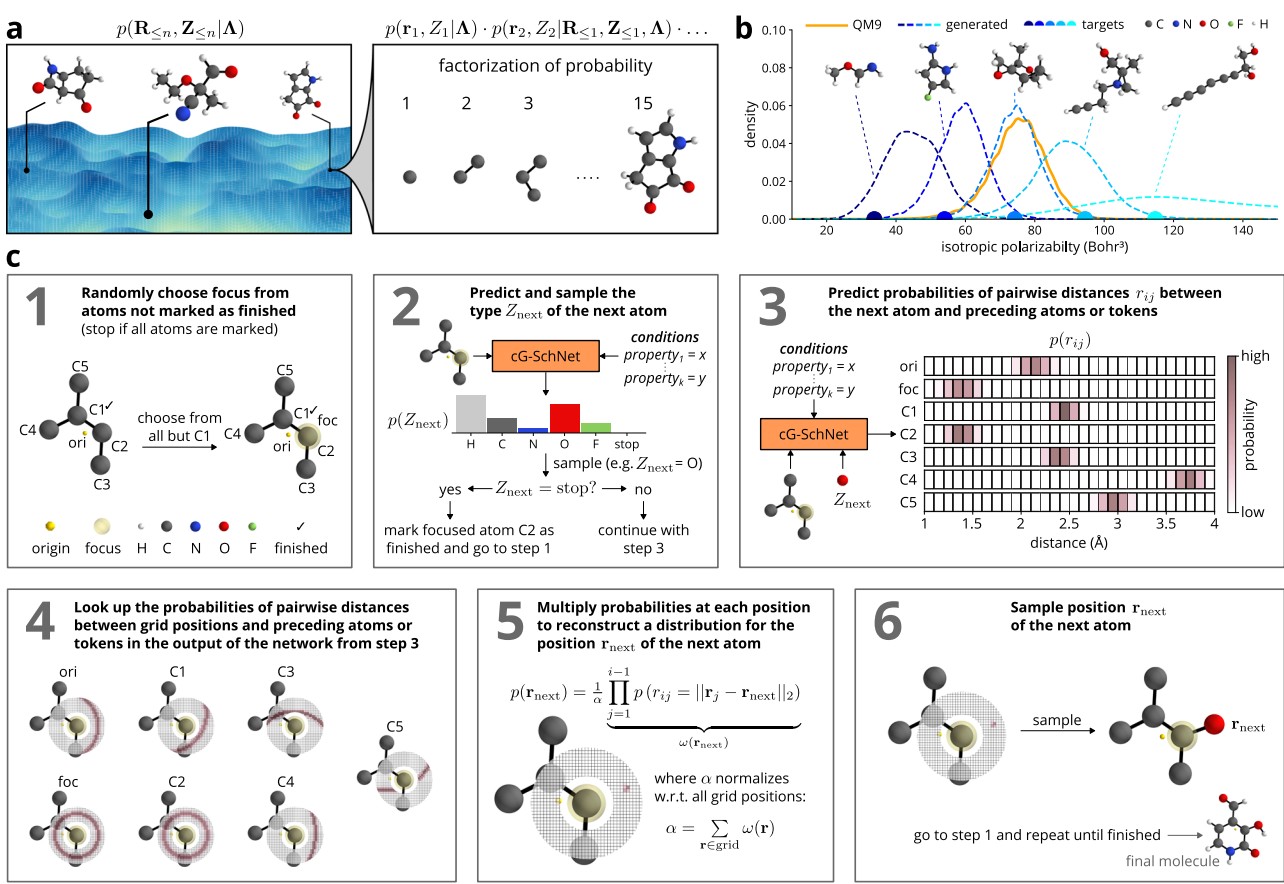

**Fig. 1 Molecule generation with cG-SchNet. a** Factorization of the conditional joint probability of atom positions and types into a chain of probabilities for placing single atoms one after another. **b** Results of sampling molecules from target-dependent conditional probability distributions. Distributions of the isotropic polarizability of training structures (orange) and five sets of molecules generated by the same cG-SchNet model (blue curves) conditioned on five different isotropic polarizability target values (color-matching dots above the *x*-axis). The generated molecule closest to the corresponding target value and not contained in the training data (unseen) is shown above each curve. **c** Schematic depiction of the atom placement loop. For visualization purposes, we show a planar molecule and a 2d slice of the actual 3d grid distributions in steps 4, 5, and 6.

shown to accurately reproduce a distribution of molecular structures[48].

Figure 2 shows a schematic depiction of the cG-SchNet architecture. The conditions $\lambda_1, \ldots, \lambda_k$ are each embedded into a latent vector space and concatenated, followed by a fully connected layer. In principle, any combination of properties can be used as conditions with our architecture with a suitable embedding network. In this work, we use three scalar-valued electronic properties such as isotropic polarizability, vector-valued molecular fingerprints, and the atomic composition of molecules. Vector-valued properties are directly processed by the network while scalar-valued targets are first expanded on a Gaussian basis. To target an atomic composition, learnable atom type embeddings are weighted by occurrence. The embedding procedure is described in detail in the Methods section.

In order to localize the atom placement and stabilize the generation procedure, cG-SchNet makes use of the same two auxiliary tokens as in the unconditional setting, namely the origin and the focus token[48]. Auxiliary tokens are treated like regular atoms by the model, i.e., they possess positions and token types, which are contained in the tuples of atom positions and atom types serving as input at each step. The origin token marks the center of the mass of molecules and allows the architecture to steer the growth from inside to outside. The focus token localizes the prediction of the next position in order to assure scalability and allows to break symmetries of partial structures. This avoids artifacts in the reconstruction of the positional distribution (Eq.

(3)) as reported by Gebauer et al.[48]. At each step, the focus token is randomly assigned to a previously placed atom. The position of the next atom is required to be close to this focus. In this way, we can use a small grid localized on the focus that does not grow with the number of atoms when predicting the distribution of the next position.

We train cG-SchNet on a set of molecular structures, where the values of properties used as conditions are known for each molecule. Given the conditions and the partial molecular structure at each step, cG-SchNet predicts a discrete distribution for the type of the next atom. As part of this, a stop type may be predicted that allows the model to control the termination of the sampling procedure and therefore generate molecules with variable size and composition. After sampling a type, cG-SchNet predicts distributions for the distance between the atom to be placed and each preceding atom and auxiliary token. The schematic depiction of the atom placement loop in Fig. 1c includes the auxiliary tokens, the model predictions, and the reconstruction of the localized 3d grid distribution. During training, we minimize the cross-entropy loss between the predicted distributions and the ground-truth distributions known from the reference calculations. For further details on the model architecture and training procedure, refer to the Methods section.

**Generating molecules with specified motifs**. In many applications, it is advantageous for molecules to possess specific functional groups or structural motifs. These can be correlated with

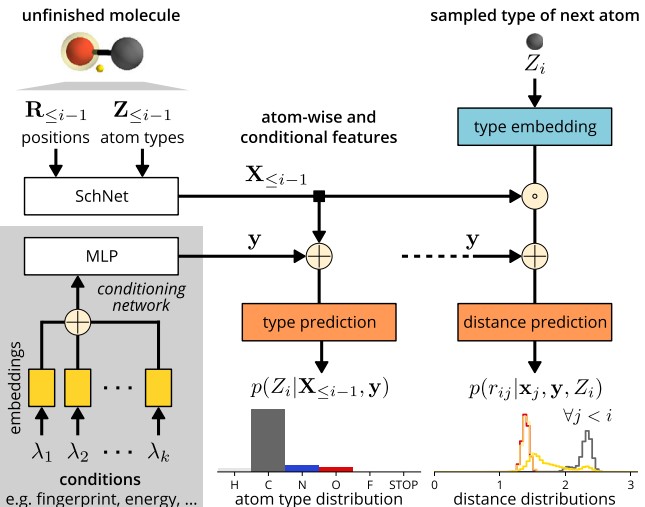

**Fig. 2 Schematic depiction of the cG-SchNet architecture with inputs and outputs.** "$\oplus$" represents concatenation and "$\odot$" represents the Hadamard product. Left: Atom-wise feature vectors representing an unfinished molecule are extracted with SchNet[67] and conditions are individually embedded and then concatenated to extract the conditional features vector. The exact embedding depends on the type of the condition (e.g., scalar or vector-valued). Middle: The distribution for the type of the next atom is predicted from the extracted feature vectors. Right: Based on the extracted feature vectors and the sampled type of the next atom, distributions for the pairwise distances between the next atom and every atom/token in the unfinished molecule are predicted. See Methods for details on the building blocks.

desirable chemical properties, e.g., polar groups that increase solubility, or with improved synthetic accessibility. In order to sample molecules with specific motifs, we condition cG-SchNet on a path-based, 1024 bits long fingerprint that checks molecular graphs for all linear segments of up to seven atoms[57] (Supplementary Methods 3). The model is trained on a randomly selected subset of 55k molecules from the QM9 dataset consisting of ~134k organic molecules with up to nine heavy atoms from carbon, nitrogen, oxygen, and fluorine[58–60]. We condition the sampling on fingerprints of unseen molecules, i.e., structures not used during training. Figure 3a shows results for four examples. We observe that the generated molecules have a higher similarity with the target fingerprints than the training data. Furthermore, structures with high target similarity are also sampled with higher probability, as can be seen from the increased similarity score of generated duplicates. In the last column of Fig. 3a, we show sampled molecules with high similarity to each target and see that in each case various structures with perfectly matching fingerprints were found. For reference, we also show the most similar molecule in the training set. Overall, we see that the conditional sampling with cG-SchNet is sensitive to the target fingerprint and allows for the generation of molecules with desired structural motifs. Although there are no molecules with the same fingerprint in the training data for three of the four fingerprint targets, the ML model successfully generates perfectly matching molecules, demonstrating its ability to generalize and explore unseen regions of chemical compound space.

**Generalization of condition-structure relationship across compositions.** For inverse design tasks, integrating information gained from different structures and properties is vital to obtain previously unknown candidates with desired properties. In this experiment, we target $C_7N_1O_1H_{11}$ isomers with HOMO-LUMO

gap values outside the range observed during training. To this end, the model has to learn from other compositions how molecules with particularly high or low HOMO-LUMO gaps are structured, and transfer this knowledge to the target composition. There are 5859 $C_7N_1O_1H_{11}$ isomers in QM9, where 997 have a HOMO-LUMO gap smaller than 6 eV, 1612 have a HOMO-LUMO gap larger than 8 eV, and 3250 lie in between these two values. We restrict the training data consisting of 55k molecules from QM9 to contain no $C_7N_1O_1H_{11}$ isomers with HOMO-LUMO gap values outside the intermediate range (Fig. 3b). Thus, the model can only learn to generate molecules with gaps outside this range from compositions other than $C_7N_1O_1H_{11}$.

Fig. 3b shows examples of generated $C_7N_1O_1H_{11}$ isomers for two target values as well as the respective HOMO-LUMO gap distributions. In both cases, the majority of generated isomers exhibit gap values close to the respective target ($\pm 1$ eV), i.e., outside of the range observed for these isomers by the model during training. This demonstrates that cG-SchNet is able to transfer knowledge about the relationship between structural patterns and HOMO-LUMO gaps learned from molecules of other compositions to generate unseen $C_7N_1O_1H_{11}$ isomers with outlying gap values upon request.

**Discovery of low-energy conformations.** The ability to sample molecules that exhibit property values that are missing in the training data is a prerequisite for the targeted exploration of chemical space. A generative model needs to fill the sparsely sampled regions of the space, effectively enhancing the available data with novel structures that show property values of interest. We study this by training cG-SchNet on a randomly sampled set of 55k QM9 molecules and query our model to sample low-energy $C_7O_2H_{10}$ isomers—the most common composition in QM9. We exclude these isomers from the training data, i.e., our model has to generalize beyond the seen compositions. The identification of low-energy conformations is desirable in many practical applications, since they tend to be more stable. However, the energy of molecules is largely determined by their size and composition. Since we are mainly interested in the energy contribution of the spatial arrangement sampled by the model, we require a normalized energy value. To this end, we define the relative atomic energy, which indicates whether the internal energy per atom is relatively high or low compared to other molecules of the same composition in the dataset (see Supplementary Methods 2 for details). Negative values indicate comparatively low energy, and thus higher stability than the average structure of this composition. Note that a similarly normalized energy has been defined by Zubatyuk et al.[61] for their neural network potential. Using the relative atomic energy allows cG-SchNet to learn the influence of the spatial arrangement of atoms on the energy and transfer this knowledge to the unseen target composition. Examples of generated $C_7O_2H_{10}$ isomers with low, intermediate, and high relative atomic energy are shown in Fig. 4a. We observe that conformations with highly strained, small rings exhibit increased relative atomic energy values.

Figure 4a shows that the trained model generalizes from the training data to sample $C_7O_2H_{10}$ isomers capturing the whole range of relative atomic energies exhibited by the QM9 test structures. We focus on stable, low-energy isomers for our analysis in the following. We sample 100k molecules with the trained cG-SchNet conditioned on the composition $C_7O_2H_{10}$ and a relative atomic energy value of $-0.1$ eV, i.e., close to the lowest energies occurring for these isomers in QM9. The generated molecules are filtered for valid and unique $C_7O_2H_{10}$ isomers, relaxed using density functional theory (DFT), and then matched with the test data structures. 169 of the 200 isomers with the

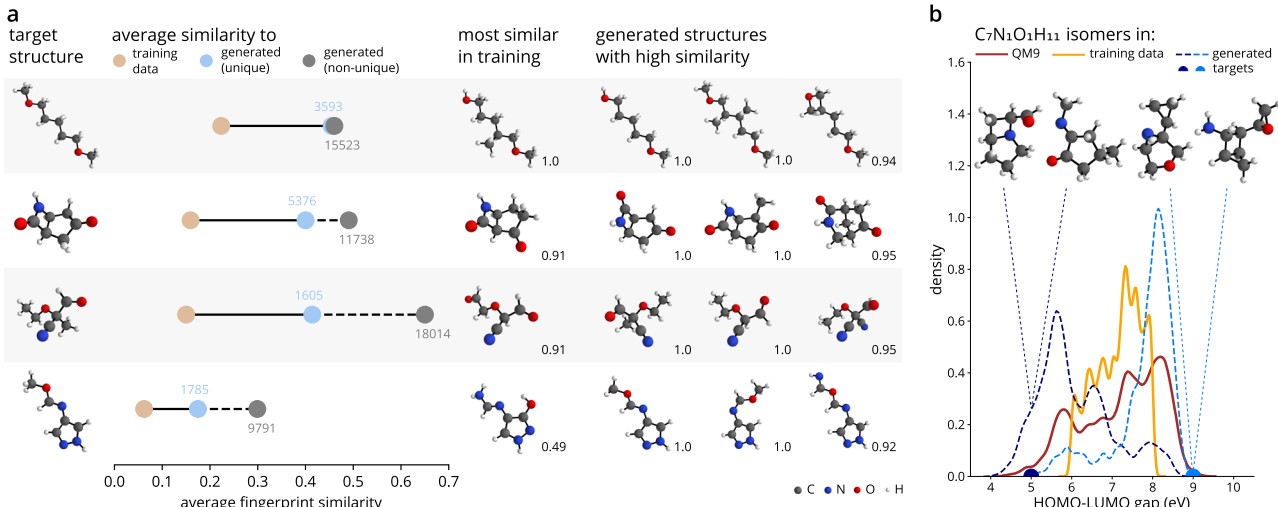

**Fig. 3 Targeted exploration of chemical space with cG-SchNet. a** Generation of molecules with desired motifs by conditioning cG-SchNet on simple path-based fingerprints. First column: Four different target fingerprints of structures from the test set. For each, we conditionally sample 20k molecules with cG-SchNet. Second column: Average Tanimoto similarity of the respective target to training structures (brown) and to generated molecules without duplicates (blue) and with duplicates (gray). The amount of generated structures is noted next to the dots. Third column: Most similar training molecule. Fourth column: Three generated unseen examples with high similarity to the target. The Tanimoto similarity to the target structure is noted to the bottom-right of depicted molecules. **b** Generation of $C_7N_1O_1H_{11}$ isomers with HOMO-LUMO gap targets outside the training data range by conditioning cG-SchNet on atomic composition and HOMO-LUMO gap. The training dataset of 55k QM9 molecules is restricted to not contain any $C_7N_1O_1H_{11}$ isomers with gap < 6 eV or gap > 8 eV. The graph shows the distribution of the gap for the $C_7N_1O_1H_{11}$ isomers in QM9 (brown), the isomers in the restricted training dataset (orange), and the two sets of isomers generated with cG-SchNet (blue curves) when targeting the composition $C_7N_1O_1H_{11}$ and two gap values outside the training data range (color-matching dots on the x-axis). For each target value, the two generated isomers closest to it are depicted.

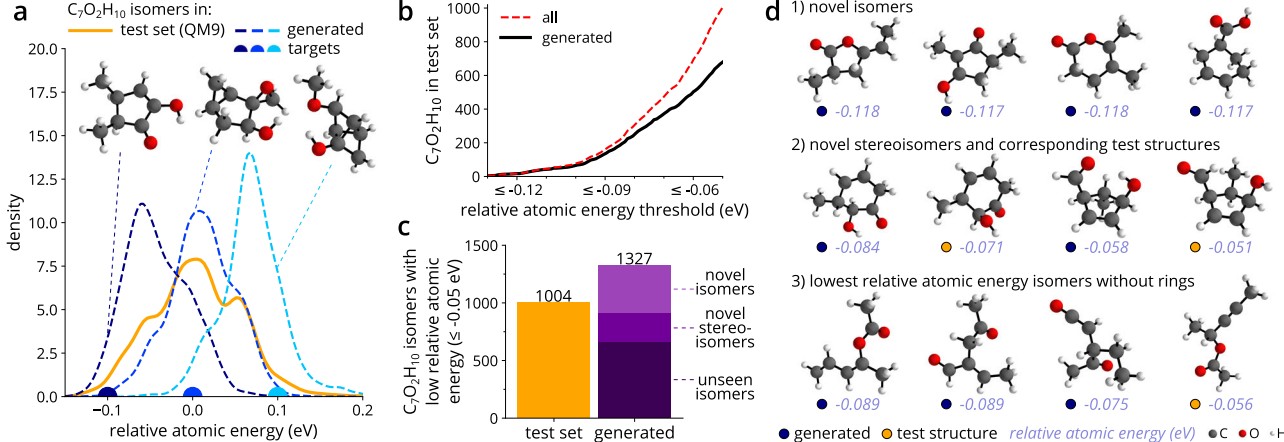

**Fig. 4 Discovery of low-energy isomers for an unseen composition.** We sample $C_7O_2H_{10}$ isomers with cG-SchNet conditioned on atomic composition and relative atomic energy (see text for details), where the training dataset was restricted to contain no $C_7O_2H_{10}$ conformations. **a** The distribution of the relative atomic energy for $C_7O_2H_{10}$ isomers in the test set (orange) and for three sets of isomers generated with cG-SchNet (blue curves) when targeting the composition $C_7O_2H_{10}$ and three different relative atomic energy values as marked with color-matching dots on the x-axis. The generated isomer closest to the respective target is depicted above each curve. **b** The absolute number of $C_7O_2H_{10}$ isomers in the test set (red dotted line) for increasing relative atomic energy thresholds. The black solid line shows how many of these were generated by cG-SchNet (target energy −0.1 eV). **c** Bar plot of the absolute number of $C_7O_2H_{10}$ isomers with relative atomic energy ≤0.05 eV in the test set (orange) and generated by cG-SchNet (target energy −0.1 eV, purple). The bar for generated molecules is divided into isomers that can be found in the test set (unseen isomers), isomers that have different stereochemistry but share the same bonding pattern as test set structures (novel stereoisomers), and novel constitutional isomers that are not in QM9 (novel isomers). **d** Relaxed example low-energy isomers generated by cG-SchNet (target energy −0.1 eV, blue dots) and structures from the test set (orange dots) along with their relative atomic energy.

lowest relative atomic energy in the test set have been recovered by the model as well as 67% of the 1k isomers with relative atomic energy lower than −0.05 eV (Fig. 4b). Beyond that, cG-SchNet has generated 416 novel isomers as well as 243 novel stereoisomers that share the same bonding pattern as a test structure but show different stereochemistry (Fig. 4c). We found 32% more unique $C_7O_2H_{10}$ isomers with relative atomic energy

lower than −0.05 eV with our model than already contained in QM9. Example isomers are depicted in Fig. 4d. For reference, we show additional, randomly selected generated novel isomers along with their most similar counterparts from QM9 in Supplementary Fig. 1 and depict how atoms in these structures moved during relaxation in Supplementary Fig. 4. These examples illustrate that cG-SchNet samples molecules that are

close to equilibrium configurations and thus require only a few steps of relaxation with DFT or a neural network potential. Furthermore, we examine different conformations found for the five most often generated isomers in Supplementary Fig. 3.

The generated molecules include structures and motifs that are sparse or not included in the QM9 benchmark dataset, which has previously been reported to suffer from decreased chemical diversity compared to real-world datasets[62]. For instance, there are no $C_7O_2H_{10}$ isomers with carboxylic acid groups in QM9, while twelve of the generated novel low-energy isomers possess this functional group (e.g., Fig. 4d, top right and Supplementary Fig. 2). Carboxylic acid groups are a common motif of organic compounds and feature prominently in fats and amino acids. While they are only contained in a few hundred molecules in QM9, cG-SchNet has learned to transfer this group to molecules of the targeted composition. Moreover, the model has discovered several acyclic $C_7O_2H_{10}$ isomers exhibiting a significantly lower relative atomic energy than those in QM9 (examples in Fig. 4d, bottom row). As cG-SchNet generalizes beyond the chemical diversity of QM9, this demonstrates that it can be employed to systematically enhance a database of molecular structures.

**Targeting multiple properties: Discovery of low-energy structures with small HOMO-LUMO gap.** For most applications, the search for suitable molecules is guided by multiple properties of interest. Therefore, a method for exploration needs to allow for the specification of several conditions at the same time. Here we demonstrate this ability by targeting HOMO-LUMO gap as well as relative atomic energy, i.e., two complex electronic properties at the same time. A particular challenging task is to find molecules with extreme property values, as those are often located at the sparsely populated borders of the training distribution. In previous work, we have biased an unconditioned G-SchNet in order to sample molecules with small HOMO-LUMO gap[48]. The model was fine-tuned with all ~3.8k available molecules from QM9 with HOMO-LUMO gap smaller than 4.5 eV, a small fraction of the whole QM9 dataset with ~130k molecules. In the following, we demonstrate that improved results can be achieved with the cG-SchNet architecture while using fewer training samples from the target region. We further condition the sampling to particularly stable, low-energy conformations. In a fine-tuning approach, this would limit the training data to only a few molecules that are both stable and exhibit small gaps. In contrast, the conditioned model is able to learn also from reference calculations where only one of the desired properties is present.

We condition cG-SchNet on the HOMO-LUMO gap as well as the relative atomic energy and train it on 55k randomly selected QM9 molecules, where only ~1.6k of the ~3.8k molecules with HOMO-LUMO gap smaller than 4.5 eV are contained. Then, we sample the same number of molecules as for the biased model[48] (20k) with the trained cG-SchNet using a HOMO-LUMO gap value of 4.0 eV and relative atomic energy of −0.2 eV as conditions. The generated conformations are filtered for valid and unique molecules, relaxed using DFT, and then matched with the training data structures.

Figure 5 compares the sets of generated, unique, unseen molecules with HOMO-LUMO gap smaller than 4.5 eV obtained for the cG-SchNet and biased G-SchNet. For biased G-SchNet, we use the previously published[48] dataset of generated molecules with a low HOMO-LUMO gap and remove all structures with HOMO-LUMO gap larger than 4.5 eV. Since the energy range has not been restricted for the biased G-SchNet, it samples structures that capture the whole space spanned by the training data, i.e., also less stable molecules with higher relative atomic energy. The molecules generated with cG-SchNet, in contrast, are

mostly structures with low relative atomic energy (Fig. 5a). Considering the total amount of unseen molecules with small gaps found by both models, we observe that cG-SchNet samples a significantly larger number of structures from the low-energy domain than the biased G-SchNet. It similarly surpasses the number of molecules from this domain in the training set, showcasing an excellent generalization performance (see Fig. 5b). For example, the model has learned to build molecules close to the target conditions that contain more than nine heavy atoms, i.e., larger than the structures from the training data. This can be seen in Supplementary Fig. 5, where we depict generated molecules with gap and relative atomic energy values beyond the training regime.

The statistics about the average atom, bond, and ring count of generated molecules depicted in Fig. 5c reveal further insights about the structural traits and differences of molecules with low HOMO-LUMO gap in the two sets. The molecules found with cG-SchNet contain more double bonds and a larger number of rings, mainly consisting of five or six atoms. This indicates a prevalence of aromatic rings and conjugated systems with alternating double and single bonds, which are important motifs in organic semiconductors. The same patterns can be found for molecules from biased G-SchNet, however, there is an increased number of nitrogen and oxygen atoms stemming from less stable motifs such as rings dominated by nitrogen. An example of this is the molecule with the highest energy depicted in Fig. 5a. Furthermore, the molecules of biased G-SchNet tend to contain highly strained small cycles of three or four atoms. cG-SchNet successfully averts these undesirable motifs when sampling molecules with a low relative atomic energy target.

We conclude that cG-SchNet has learned to build stable molecules with a low HOMO-LUMO gap even though it has seen less than half of the structures that the biased model was fine-tuned on. More importantly, the training data contains only very few (~200) structures close to the target conditions at the border of the QM9 distribution, i.e., with HOMO-LUMO gap smaller than 4.5 eV and relative atomic energy smaller than −0.1 eV. However, our model is able to leverage information even from structures where one of the properties is outside the targeted range. Consequently, it is able to sample a significantly higher number of unseen molecules from the target domain than there are structures in the training data that fulfill both targets. In this way, multiple properties can be targeted at once in order to efficiently explore chemical compound space.

The efficiency of cG-SchNet in finding molecular structures close to the target conditions is particularly evident compared to an exhaustive enumeration of graphs with subsequent relaxation using DFT. In both cases, the relaxation required to obtain equilibrium coordinates and the physical properties is the computational bottleneck and takes more than 15 min per structure for the molecules generated in this experiment. Furthermore, the calculation of the internal energy at zero Kelvin (U0) requires additional 40 min per molecule. In contrast, the generation with cG-SchNet takes only 9 ms per structure on a Nvidia A100 GPU when sampling in batches of 1250. The training time of about 40 hours is negligible, as it corresponds to the relaxation and calculation of U0 of only 44 structures. Thus, the efficiency is determined by the number of molecules that need to be relaxed for each method. The QM9 dataset was assembled by relaxing structures from the GDB enumeration[60] of graphs for small organic compounds. Of the ~78k molecules that we did not use for training, 354 molecules are close to the target region. Relaxing only the 5283 structures proposed by cG-SchNet, i.e., less than 10% of the computations performed by screening all graphs, we can already recover 46% of these structures. Additionally, the model has unveiled valid molecules close to

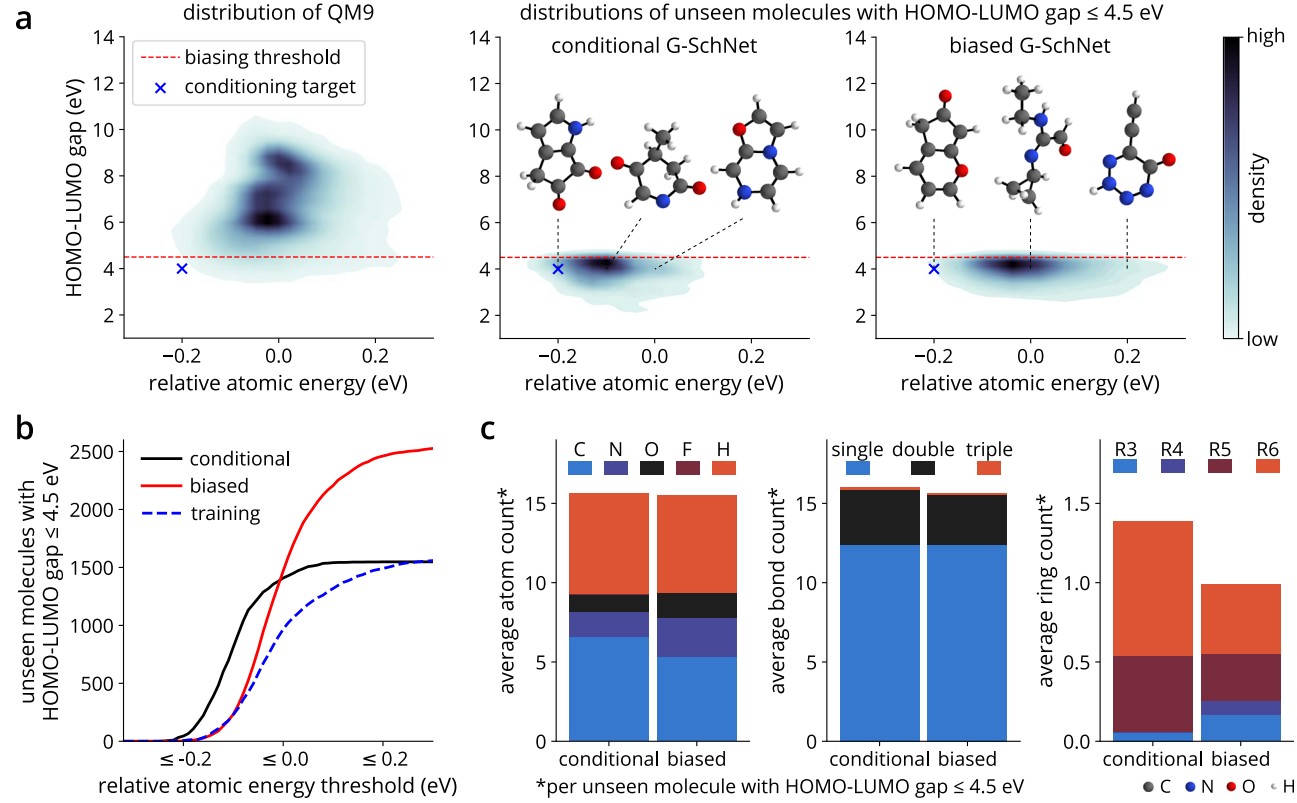

**Fig. 5 Discovery of low-energy structures with small HOMO-LUMO gap.** We compare cG-SchNet to the previous, biased G-SchNet approach[48]. **a** The joint distributions of relative atomic energy and HOMO-LUMO gap for QM9 (left) and for unique, unseen molecules with gap ≤4.5 eV generated with cG-SchNet (middle) and with biased G-SchNet (right). Biased G-SchNet was fine-tuned on all molecules in QM9 below a gap threshold of 4.5 eV (red, dotted line). The conditions used for generation with cG-SchNet are marked with a blue cross. The depicted molecules are generated examples with a gap of 4 eV and different relative atomic energy values (black, dotted lines). More examples as well as the distributions close to the conditioning target for cG-SchNet and the training data can be found in Supplementary Fig. 5. **b** The absolute number of unique, unseen molecules with gap ≤4.5 eV generated by cG-SchNet (black) and biased G-SchNet (red) for increasing relative atomic energy thresholds. For reference, we also show the amount of structures with low gap included in the training set of cG-SchNet (blue dotted line). **c** The average number of atoms of different types (left), bonds of different orders (middle), and rings of different sizes (right) in unique, unseen molecules with gap ≤4.5 eV generated by each model.

the target that are not contained in the dataset. More than 380 of these are larger than QM9 structures and thus not covered. However, 253 smaller structures were missed by the enumeration method. This is, again, in line with findings by Glavatskikh et al.[62] that even for these small compounds the graph-based sampling does not cover all structures of interest. Consequently, we obtain more than two times the amount of molecules close to the target property values with cG-SchNet than with the exhaustive enumeration method while requiring less than 10% of the computation time.

The conditional model is not restricted to the space of low-energy / low gap molecules, but can also sample low-energy / high gap structures or any other combination of interest. Thus, the efficiency of the generative model becomes even more pronounced when there are multiple sets of desirable target values. Figure 1b depicts an example where cG-SchNet has been trained on the isotropic polarizability as a condition. Here, the same model is employed to sample molecules for five different target values. Again, cG-SchNet is able to generalize to isotropic polarizabilities beyond the values present in the training data.

## Discussion

cG-SchNet enables the targeted discovery of 3d molecular structures conditioned on arbitrary combinations of multiple structural and chemical properties. The neural network captures global and local symmetries of molecular structures by design, enabling it to learn

complex relationships between chemical properties and 3d structures. This makes it possible to generalize to unseen conditions and structures, as we have thoroughly evaluated in a line of experiments where we target property values not included in the training data. In contrast to previous approaches, the model does not require target-specific biasing procedures. Instead, the explicit conditioning enables cG-SchNet to learn efficiently from all available reference calculations. Desirable values of multiple properties can be targeted simultaneously to sample from specific conditional distributions. In this way, cG-SchNet generates novel 3d candidate molecules that exhibit the target properties with high probability and thus are perfectly suited for further filtering and evaluation using ML force fields.

Further work is required to apply the cG-SchNet architecture to the exploration of significantly larger systems and a more diverse set of atom types. Although an unconditional G-SchNet has been trained on drug-like molecules with 50+ atoms in the 3D-Scaffold framework[54], adjustments will be necessary to ensure scalability to materials. In the current implementation, we employ all preceding atoms to predict the type and reconstruct the positional distribution of the next atom. Here, a cutoff or other heuristics to limit the number of considered atoms will need to be introduced, together with corrections for long-range interactions. While the small organic compounds considered in this work are well represented by QM9, the model might benefit from enhancing the training data using representative building blocks such as "amons"[63] or other fragmentation methods[64,65]. This becomes

increasingly important when tackling larger molecules where reference data is hard to obtain. Another direction for future work is the extended comparison of cG-SchNet to established methods in different fields, e.g., for the discovery of drugs or materials, to identify promising applications and possible shortcomings. Furthermore, additional adaptations are necessary to explore systems with periodic boundary conditions. In cases where not all targeted properties can be fulfilled simultaneously, finding suitable molecules becomes harder, if not impossible. Therefore, another important extension is to explicitly define a trade-off between multiple conditions or to sample along a Pareto front.

We have applied cG-SchNet to sample particularly stable, low-energy $C_7O_2H_{10}$ isomers. In this process, we have discovered molecules and motifs that are absent from the QM9 database, such as isomers with carboxylic acid groups. Furthermore, we have sampled more than 800 low-energy molecules with HOMO-LUMO gaps smaller than 4.5 eV from a domain that is only sparsely represented in the training data. Although the exploration of such small molecules with an exhaustive sampling of molecular graphs and subsequent evaluation with DFT is computationally feasible, our model considerably accelerates the process by providing reasonable candidate structures. cG-SchNet thus also enables the data-efficient, systematic improvement of chemical databases, which is particularly valuable considering the computational cost and unfavourable scaling of electronic structure calculations. This paves the way for ML-driven, targeted exploration of chemical compound space and opens avenues for further development towards generative models for larger and more general atomistic systems.

## Methods

**Training data.** For each training run, 55k reference structures are randomly sampled from the QM9 dataset[58–60], a collection of 133,885 molecules with up to nine heavy atoms from carbon, nitrogen, oxygen, and fluorine. We removed 915 molecules from the training pool which are deemed invalid by our validation procedure that checks the valency and connectedness of generated structures (see Section Checking validity and uniqueness of generated molecules). For some runs, limited subsets of the training data pool are used, as described in the results (e.g., without $C_7O_2H_{10}$ isomers). We train the neural network using 50k randomly sampled molecules and employ the remaining 5k for validation (see Section Neural network training). All molecules shown in figures have been rendered with the 3d visualization package Mayavi[66].

**Details on the neural network architecture.** In the following, we describe the cG-SchNet architecture as depicted in Figure 2 in detail. We use the shifted softplus non-linearity

$$\text{ssp}(x) = \ln\left(\frac{1}{2}e^x + \frac{1}{2}\right) \tag{4}$$

throughout the architecture. Successive linear neural network layers with intermediate shifted softplus activation are written as

$$\text{mlp}(\mathbf{x}) = \mathbf{W}_2^T \text{ssp}\left(\mathbf{W}_1^T \mathbf{x} + \mathbf{b}_1\right) + \mathbf{b}_2 \tag{5}$$

with input $\mathbf{x} \in \mathbb{R}^{n_{in_1}}$, weights $\mathbf{W}_1 \in \mathbb{R}^{n_{in_1} \times n_{in_2}}$, $\mathbf{W}_2 \in \mathbb{R}^{n_{in_2} \times n_{out}}$, and biases $\mathbf{b}_1 \in \mathbb{R}^{n_{in_2}}$, $\mathbf{b}_2 \in \mathbb{R}^{n_{out}}$. While this example shows a succession of two linear layers, the notation covers any number of successive linear layers with intermediate shifted softplus activations in the following. The number of layers and neurons as well as all other hyper-parameter choices for our neural network architecture are given in Supplementary Table 1.

The inputs to cG-SchNet when placing atom $i$ is a partial molecule consisting of $i-1$ atoms including two auxiliary tokens (focus and origin) and $k$ target properties $\Lambda = (\lambda_1, ..., \lambda_k)$. The atoms and tokens are given as tuples of positions $\mathbf{R}_{\leq i-1} = (\mathbf{r}_1, ..., \mathbf{r}_{i-1})$ with $\mathbf{r}_j \in \mathbb{R}^3$ and types $\mathbf{Z}_{\leq i-1} = (Z_1, ..., Z_{i-1})$ with $Z_j \in \mathbb{N}$. The first two entries correspond to the auxiliary tokens, which are treated like ordinary atoms by the neural network. Thus, whenever we refer to atoms in the following, this also encompasses the tokens. Note that tokens do not influence the sampling probability of a molecule in Eq. (1), since they are placed with probability $p(\mathbf{R}_{\leq 2}, \mathbf{Z}_{\leq 2} | \Lambda) = 1$.

We employ SchNet[21,67] to extract atom-wise features $\mathbf{X}_{\leq i-1} = (\mathbf{x}_1, ..., \mathbf{x}_{i-1})$ that are invariant to rotation and translation. We use the SchNet representation

network as implemented in the SchNetPack software package[68] with $F = 128$ features per atom and 9 interaction blocks.

Additionally, we construct a vector $\mathbf{y} \in \mathbb{R}^D$ of conditional features from the list of target properties. To this end, each target property is first mapped into vector space using an individual embedding network that depends on the form of the specific property. In this work, we employ different embedding networks for scalar-valued properties, vector-valued properties, and atomic composition. Scalar-valued properties are processed by an MLP after applying a Gaussian radial basis function expansion

$$\mathbf{f}_{\text{scal}} = \text{mlp}\left(\left[e^{-\frac{(\lambda_{\text{scal}} - (\lambda_{\min} + l\Delta\omega))^2}{2\Delta\omega^2}}\right]_{0 \leq l \leq \frac{\lambda_{\max} - \lambda_{\min}}{\Delta\omega}}\right) \tag{6}$$

where the minimum $\lambda_{\min}$ and maximum $\lambda_{\max}$ property values and the grid spacing $\Delta\omega$ are hyper-parameters chosen per target property. Vector-valued properties such as molecular fingerprints are directly processed by an MLP:

$$\mathbf{f}_{\text{vec}} = \text{mlp}\left(\lambda_{\text{vec}}\right). \tag{7}$$

For the atomic composition, we use two embedding blocks. While the number of atoms is embedded as a scalar property, we map atom types to learnable embeddings $\mathbf{g}_Z^{\text{comp}} \in \mathbb{R}^G$. These vectors are weighted by the fraction of the corresponding atom type in the target atomic composition, concatenated, and processed by an MLP. For example, the atomic composition of hydrocarbons would be encoded as:

$$\mathbf{f}_{\text{comp}} = \text{mlp}\left(\left[n_H\,\mathbf{g}_H^{\text{comp}} \oplus n_C\,\mathbf{g}_C^{\text{comp}}\right]\right) \tag{8}$$

where "$\oplus$" is the concatenation of two vectors and $n_H$ and $n_C$ is the fraction of hydrogen and carbon atoms in the target atomic composition, respectively. Finally, the property feature vectors $\mathbf{f}_{\lambda_1}, ..., \mathbf{f}_{\lambda_k}$ are aggregated by an MLP

$$\mathbf{y} = \text{mlp}\left(\left[\mathbf{f}_{\lambda_1} \oplus \mathbf{f}_{\lambda_2} \oplus ... \mathbf{f}_{\lambda_k}\right]\right), \tag{9}$$

to obtain the combined conditional features $\mathbf{y}$.

Given the conditional features $\mathbf{y}$ representing the target properties and the atom-wise features $\mathbf{X}_{\leq i-1}$ describing the partial molecule, the cG-SchNet architecture predicts distributions for the type of the next atom and its pairwise distances to all preceding atoms with two output networks. Let $\mathcal{Z}^{\text{all}} \subset \mathbb{N}$ be the set of all atom types in the training data including an additional stop marker type. The type prediction network first computes atom-wise, $|\mathcal{Z}^{\text{all}}|$-sized vectors

$$\mathbf{s}_j = \text{mlp}\left(\left[\mathbf{x}_j \oplus \mathbf{y}\right]\right) \quad \text{with } j < i \tag{10}$$

containing a scalar score for each atom type. Let $\mathbf{s}_j^{[z]}$ be the score of type $z \in \mathcal{Z}^{\text{all}}$ predicted for preceding atom $j$. Then, the probability for the next atom being of type $z$ is obtained by taking the softmax over all types and averaging the atom-wise predictions:

$$p(Z_i = z | \mathbf{X}_{\leq i-1}, \mathbf{y}) = \frac{1}{i-1}\sum_{j=1}^{i-1} \frac{e^{\mathbf{s}_j^{[z]}}}{\sum_{z' \in \mathcal{Z}^{\text{all}}} e^{\mathbf{s}_j^{[z']}}}. \tag{11}$$

The distance distributions are discretized on a grid with $L$ bins, each covering a span of $\Delta\mu$. The bin of a distance $d \in \mathbb{R}^+$ is given by $b : \mathbb{R}^+ \mapsto \{1, ..., L\}$

$$b(d) = \begin{cases} \left\lceil\frac{d + \frac{1}{2}\Delta\mu}{\Delta\mu}\right\rceil & \text{if } d \leq (L-1)\Delta\mu \\ L & \text{if } d > (L-1)\Delta\mu \end{cases}. \tag{12}$$

Given the type $Z_i$ of the next atom, the distance prediction network computes scores for each preceding atom and distance bin

$$\mathbf{u}_j = \text{mlp}\left(\left[\left(\mathbf{x}_j \odot \mathbf{g}_{Z_i}^{\text{next}}\right) \oplus \mathbf{y}\right]\right) \quad \forall j < i \tag{13}$$

where "$\odot$" is the Hadamard product and $\mathbf{g}_Z^{\text{next}} \in \mathbb{R}^F$ is a learnable atom type embedding. The probability of any distance between the new atom and a preceding atom is obtained by applying a softmax over all bins

$$p(r_{ij} = d | \mathbf{x}_j, \mathbf{y}, Z_i) = \frac{e^{\mathbf{u}_j^{[b(d)]}}}{\sum_{l=1}^{L} e^{\mathbf{u}_j^{[l]}}} \quad \forall j < i \tag{14}$$

where $\mathbf{u}_j^{[b(d)]}$ is the score of bin $b(d)$ predicted for preceding atom $j$.

**Sampling atom placement sequences for training.** The number of sequences in which a molecule can be built by placing $n$ atoms grows factorially with $n$. During training, we randomly sample a new atom placement sequence for every training molecule in each epoch. However, we use the focus and origin tokens to constrain how molecules are built by cG-SchNet and thus significantly reduce the number of possible sequences. Our approach ensures that molecules tend to grow outwards starting from the center of mass and that each new atom is placed close to one of the already placed atoms. For the first atom placement step, we set the positions of the focus and origin tokens to the center of mass of the training molecule and

choose the atom closest to it as the first atom to be placed. If multiple atoms are equally close, one of them is randomly chosen as the first atom.

Afterwards, each atom placement step follows the same procedure. One of the already placed atoms (excluding tokens) is chosen as focus, i.e., the position of the focus token is set to the position of the chosen atom. Then, from all unplaced atoms, we select the neighbor of the focus that is closest to the center of mass as the next atom. If there are no neighbors of the focus among the unplaced atoms, we insert a step where the type prediction network shall predict the stop marker type. In this way, the focus atom is marked as finished before randomly choosing a new focus and proceeding with the next atom placement step. Marked atoms cannot be chosen as focus anymore and the atom placement sequence is complete when all placed atoms are marked as finished. Thus, the sequence ends up with $2n$ steps, as each atom needs to be placed and furthermore marked as finished.

For our experiments, we consider atoms sharing a bond as neighbors. However, note that bonding information is not necessarily required as neighborhood can also be defined by a radial cutoff of, e.g., 3 Å centered on the focus atom.

**Neural network training**. We use mini-batches with $M$ molecules for training. Each mini-batch contains one atom placement sequence per molecule, randomly sampled in each epoch as explained in Section Sampling atom placement sequences for training. Each step of the atom placement sequence $a \in \mathcal{A}_m$ consists of types $\mathbf{Z}_{\leq i-1}$ and positions $\mathbf{R}_{\leq i-1}$ of already placed atoms and the two auxiliary tokens, of the values $\Lambda$ of molecule $m$ for the target properties of the model, and of the type $Z_{\text{next}}$ and position $\mathbf{r}_{\text{next}}$ of the next atom.

For each atom placement, we minimize the cross-entropy between the distributions predicted by the model given $\mathbf{Z}_{\leq i-1}$, $\mathbf{R}_{\leq i-1}$, and $\Lambda$ and the distributions obtained from the ground-truth next type $Z_{\text{next}}$ and position $\mathbf{r}_{\text{next}}$. The ground-truth distribution of the next type is a one-hot encoding of $Z_{\text{next}}$, thus the cross-entropy loss for the type distributions is

$$\ell^{\text{type}}(a) = -\log\big(p(Z_i = Z_{\text{next}}|\mathbf{X}_{\leq i-1}, \mathbf{y})\big). \quad (15)$$

The average cross-entropy loss for the distance distributions is

$$\ell^{\text{dist}}(a) = -\frac{1}{i-1}\sum_{j=1}^{i-1}\sum_{l=0}^{L-1}\mathbf{q}_{jl}^{\text{next}}\log\big(\mathbf{p}_{jl}^{\text{next}}\big) \quad (16)$$

with model predictions

$$\mathbf{p}_{jl}^{\text{next}} = p\big(r_{ij} = l\Delta\mu|\mathbf{x}_j, \mathbf{y}, Z_i\big) \quad (17)$$

and Gaussian expanded ground-truth distance

$$\mathbf{q}_{jl}^{\text{next}} = \frac{e^{-\gamma(||\mathbf{r}_{\text{next}}-\mathbf{r}_j||_2 - l\Delta\mu)^2}}{\sum_{l'=0}^{L-1}e^{-\gamma(||\mathbf{r}_{\text{next}}-\mathbf{r}_j||_2 - l'\Delta\mu)^2}} \quad (18)$$

where $L$ is the number of bins of the distance probability grid with spacing $\Delta\mu$. The width of the Gaussian expansion can be tuned with $\gamma$, which we set to $\frac{10}{\Delta\mu}$ in our experiments.

The loss for a mini-batch $C$ is the average type and distance loss of all atom placement steps of all $M$ molecules in the mini-batch:

$$\ell(C) = \frac{1}{M}\sum_{m=1}^{M}\sum_{a \in \mathcal{A}_m}\left(\frac{\ell^{\text{type}}(a)}{|\mathcal{A}_m|} + \frac{\delta(a)\ell^{\text{dist}}(a)}{0.5|\mathcal{A}_m|}\right) \quad (19)$$

where $|\mathcal{A}_m|$ is the number of steps in sequence $\mathcal{A}_m$ and

$$\delta(a) = \begin{cases} 0 & \text{if } Z_{\text{next}} = \text{STOP} \\ 1 & \text{else}. \end{cases} \quad (20)$$

The indicator function $\delta$ is zero for steps where the type to predict is the stop marker, since no position is predicted in these steps.

The neural networks were trained with stochastic gradient descent using the ADAM optimizer[69]. We start with a learning rate $\eta = 10^{-4}$ which is reduced using a decay factor of 0.5 after 10 epochs without improvement of the validation loss. The training is stopped at $\eta \leq 10^{-6}$. We use mini-batches of 5 molecules and the model with the lowest validation error is selected for generation.

**Conditional generation of molecules**. For the generation of molecules, conditions need to be specified covering all target properties the model was trained on, e.g., the atomic composition and the relative atomic energy. The generation is an iterative process where the type and position of each atom are sampled sequentially using the distributions predicted by cG-SchNet. Generating a molecule with $n$ atoms takes $2n$ steps, as each atom needs to be placed and furthermore marked as finished in order to terminate the generation process.

At each step, we want to sample the type $Z_{\text{next}} \in \mathcal{Z}^{\text{all}} \subset \mathbb{N}$ and position $\mathbf{r}_{\text{next}} \in \mathbf{G} \subset \mathbb{R}^3$ of the next atom given the types and positions of already placed atoms (including the two tokens) and the conditions. Here, $\mathcal{Z}^{\text{all}}$ is the set of all atom types in the training data including an additional stop marker type and $\mathbf{G}$ is a grid of candidate positions in 3d space (see Supplementary Methods 1). An unfinished atom is randomly chosen as focus at the start of each step, i.e., the position of the focus token is aligned with the position of the chosen atom. Then, we predict the distribution of the type of the next atom with the model (see Eq. (11)) to sample

the next type

$$Z_{\text{next}} \sim p(Z_i = Z_{\text{next}}|\mathbf{X}_{\leq i-1}, \mathbf{y}). \quad (21)$$

If the next type is the stop marker, we mark the currently focused atom as finished and proceed with the next step by choosing a new focus without sampling a position. Otherwise, we proceed to predict the distance distributions between placed atoms and the next atom with the model (see Eq. (14)). Since cG-SchNet is trained to place atoms in close proximity to the focused atom, we align the local grid of candidate positions with the focus at each step regardless of the number of atoms in the unfinished molecule. Then, the distance probabilities are aggregated to compute the distribution over 3d candidate positions in the proximity of the focus. The position of the next atom is drawn accordingly

$$\mathbf{r}'_{\text{next}} \sim \frac{1}{\alpha}\prod_{j=1}^{i-1}p\big(r_{ij} = ||\mathbf{r}_j - \mathbf{r}'_{\text{next}}||_2|\mathbf{x}_j, \mathbf{y}, Z_i\big) \quad (22)$$

with

$$\mathbf{r}'_{\text{next}} = \mathbf{r}_{\text{next}} + \mathbf{r}_{\text{focus}} \quad (23)$$

where $\alpha$ is the normalization constant and $\mathbf{r}_{\text{focus}}$ is the position of the focus token. At the very first atom placement step, we center the focus and grid on the origin token, while for the remaining steps, only atoms will be focused.

The generation process terminates when all regular atoms have been marked as finished. In this work, we limit the model to a maximum number of 35 atoms. If the model attempts to place more atoms, the generation terminates and the molecule is marked as invalid.

**Checking validity and uniqueness of generated molecules**. We use Open Babel[57] to assess the validity of generated molecules. Open Babel assigns bonds and bond orders between atoms to translate the generated 3d representation of atom positions and types into a molecular graph. We check if the valence constraints hold for all atoms in the molecular graph and mark the molecule as invalid if not. Furthermore, the generated structure is considered invalid if it consists of multiple disconnected graphs. We found that Open Babel may struggle to assign correct bond orders even for training molecules if they contain aromatic sub-structures made of nitrogen and carbon. Thus, we use the same custom heuristic as in previous G-SchNet work[48] that catches these cases and checks whether a correct bond order can be found. The corresponding code is publicly available (see Code availability).

The uniqueness of generated molecules is checked using their canonical SMILES[30] string representation obtained from the molecular graph with Open Babel. If two molecules share the same string, they are considered to be equal, i.e., non-unique. Furthermore, we check the canonical SMILES string of mirror images of generated structures, which means that mirror-image stereoisomers (enantiomers) are considered to be the same molecule in our statistics. In case of duplicates, we keep the molecule sampled first, with the exception of the search for $C_7O_2H_{10}$ isomers, where we keep the structure with the lowest predicted relative atomic energy. Molecules from the training and test data are matched with generated structures in the same way, using their canonical SMILES representations obtained with Open Babel and the custom heuristic for bond order assignment. In general, we use isomeric SMILES strings that encode information about the stereochemistry of 3d structures. Only in the search for $C_7O_2H_{10}$ isomers, we also compare non-isomeric canonical SMILES obtained with RDKit[70] in order to identify novel stereoisomers, i.e., structures that share the same non-isomeric SMILES representation but differ in the isomeric variant.

**Prediction of property values of generated molecules**. We use pretrained SchNet[21,67] models from SchNetPack[68] to predict the HOMO-LUMO gap, iso-tropic polarizability, and internal energy at zero Kelvin of generated molecules. The reported mean absolute error (MAE) of these models is 0.074 eV, 0.124 Bohr$^3$, and 0.012 eV, respectively. The predicted values are used to plot the distributions of the respective property in Fig. 1b, Fig. 3b, and Fig. 4a. We relax generated molecules for every experiment in order to assess how close they are to equilibrium configura-tions and to calculate the MAE between predictions for generated, unrelaxed structures and the computed ground-truth property value of the relaxed structure. The relaxation procedure is described in Supplementary Methods 4, where fur-thermore a table with the results can be found (Supplementary Table 2). For the statistics depicted in Fig. 4b-d and Fig. 5, we use the property values computed during relaxation instead of predictions from SchNet models.

## Data availability

The molecules generated with cG-SchNet are available at www.github.com/atomistic-machine-learning/cG-SchNet(DOI 10.5281/zenodo.5907027[71]). The QM9 dataset is available under DOI 10.6084/m9.figshare.978904. The set of molecules with small HOMO-LUMO gap generated by biased G-SchNet is available at http://quantum-machine.org/datasets.

## Code availability

The code for cG-SchNet is available at www.github.com/atomistic-machine-learning/cG-SchNet(DOI 10.5281/zenodo.5907027[71]). This includes the routines for training and

deploying the model, for filtering generated structures, all hyper-parameter settings used in our experiments, and the splits of the data employed to train the reported models.

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

## Acknowledgements

N.W.A.G. and M.G. work at the BASLEARN—TU Berlin/BASF Joint Lab for Machine Learning, co-financed by TU Berlin and BASF SE. N.W.A.G., K.T.S., S.S.P.H., and K.R.M. acknowledge support by the Federal Ministry of Education and Research (BMBF) for the Berlin Institute for the Foundations of Learning and Data (BIFOLD) (01IS18037A).

K.R.M. acknowledges financial support under the Grants 01IS14013A-E, 01GQ1115, and 01GQ0850; Deutsche Forschungsgemeinschaft (DFG) under Grant Math+, EXC 2046/1, Project ID 390685689 and K.R.M. was partly supported by the Institute of Information & Communications Technology Planning & Evaluation (IITP) grants funded by the Korea Government (No. 2019-0-00079, Artificial Intelligence Graduate School Program, Korea University).

## Author contributions

N.W.A.G. developed the method and carried out the experiments. M.G. carried out the reference computations and simulations. S.S.P.H. trained the neural networks for predictions of molecular properties. N.W.A.G., M.G., K.R.M., and K.T.S. designed the experiments and analyses. N.W.A.G., M.G., and K.T.S. wrote the paper. All authors discussed results and contributed to the final version of the manuscript.

## Funding

## Competing interests

The authors declare no competing interests.
