## [Peer Review File · Nature Communications]

REVIEWER COMMENTS

Reviewer #1 (Remarks to the Author):

Summary:

The authors propose a conditional generative model for 3d molecular structures cG-SchNet. The model is conditioned on specified structural and chemical properties and the generated samples tend to have such properties in practice. The utility of the method is illustrated in several example problems where new molecules are generated with specific motifs, low energy values, and jointly targeting multiple electronic properties beyond the training regime.

What are the noteworthy results?

The main noteworthy result is the successful extension of a previous unconditional generative model of 3D molecular structures, G-SchNet, to the conditional setting. Previously, targeted generation with G-SchNet required to bias the method by fine-tuning it on a fraction of the training data set with desired properties. This created problems because usually interesting regions for exploration are often those where data is sparse. The new conditional model learns conditional distributions depending on structural or chemical properties and should in principle be less affected by that problem. Also cG-SchNet can jointly target multiple properties without the need to retrain or constrain the sampling process. cG-SchNet makes more efficient use of data resulting in increased generalization.

Will the work be of significance to the field and related fields?

The improvements over G-SchNet are significant and the field will be interested in this work. It seems to me that this is one of the first works to address the problem of conditional generation of 3D structures given properties of interests and, therefore, a significant contribution to the field.

How does it compare to the established literature? If the work is not original, please provide relevant references.

The experimental comparisons are mainly with respect to G-SchNet and the gains seem significant. There are no comparisons to other existing methodology, probably because other approaches do not work by conditioning on properties but instead optimize a reward that could balance different desired properties, including the energy of the resulting molecular configuration. The limitation of the proposed method is that it needs a dataset of 3D structures to be trained, while the reinforcement learning based methods such as [51,52] do not.

Does the work support the conclusions and claims, or is additional evidence needed?

The experiments show that cG-SchNet can sample particularly stable, low-energy $C_7O_2H_{10}$ isomers discovering molecules and motifs that are absent from the QM9 database. cG-SchNet is also used to sample low-energy molecules with small HOMO-LUMO gaps from a domain that is only sparsely represented in the training data. These experiments are enough to show that the proposed method seems to work well in practice.

Are there any flaws in the data analysis, interpretation and conclusions? Do these prohibit publication or require revision?

I did not spot any flaws in the paper. I would encourage the authors to address my comments on the clarity of the molecule generation process as described below.

Is the methodology sound? Does the work meet the expected standards in your field?

The methods section describes sound methodology. The authors mention at the end a limitation of the G-SchNet-style approach, which has a cost that is quadratic in the number of atoms. This will prevent the method from working well in large molecules. Another limitation seems the discretization used to sample the atom locations. It is unclear how this can affect performance in practice.

Is there enough detail provided in the methods for the work to be reproduced?

I found the description of the sampling process to be confusing. Especially the part on the conditional generation of molecules in appendix E. From the paper, it is hard to understand what is the connection between equation 22 and equation 14. It seems that the authors multiply all the

probabilities for each particular grid cell as given by conditioning on all past atoms and then normalize. What is the justification for this?

I think the paper would benefit from having a figure showing a cartoon description of how the molecule generation process works. The figure could include several vignettes corresponding to different steps of the process.

Some minor comments:

The authors should verify that references to arxiv papers cite instead the actual published source.

Figure 1 b -> c

isomers exhibits HOMO-LUMO -> isomers exhibit HOMO-LUMO

Reviewer #2 (Remarks to the Author):

The manuscript presents a generative neural network architecture for 3D molecular structures that can be biased towards specific chemical properties and molecular motifs. The authors demonstrate that the model can generate new organic molecules biased towards specific atomization energies, motifs, and other chemical properties.

This manuscript is well written and would be valuable to the scientific community as is. However, additional experiments involving the interactions highlighted by the authors in the introduction to be challenging to solve without a coordinate-based model (long-range interactions, conformers, metallic complexes, etc.) would significantly strengthen this work. The current experiments only involve relatively simple organic chemistry and can be done with a graph or smiles-based generative model. Aside from this concern, I would recommend this work for publication in Nature Communications.

Reviewer #3 (Remarks to the Author):

Review of Gebauer et al.

This manuscript presents a machine learning-based approach to generate molecular structures with given stoichiometry that exhibit particular properties. The ansatz pursued is that of a conditional generative neural network.

Detailed comments:

1. It appears that the ansatz pursued assumes that the emergence of a chemical property, e.g. the HOMO-LUMO gap, is a smooth function of chemical composition and molecular structure. This, however, is not necessarily the case. It would be of interest to report the HOMO-LUMO gap along such a "chemical trajectory" as indicated on by the construction procedure in eq. (2) and see how the target property "develops" or "emerges" as the molecule grows. Or does the model just need to "cover" the desired range of observable with given chemistry?
2. P. 4 states that "..Thus, the model can only learn to generate molecules with gaps outside this range from other compositions." What are these "other compositions" in the present case?
3. The results suggest that the model increases the probability to generate molecular structures and chemical compositions that contain

the desired target property. The question is whether the procedure proposed speeds up discovery relative to exhaustive sampling of all graphs. Given that DFT calculations are computationally inexpensive it may well be that generating the ML-based model, training it, evaluating and testing it is considerably more expensive than straightforward generation of the necessary graphs and evaluating and filtering for the desired property. A careful discussion of this question is required and where the "break-even point" for given cost/benefit is. In other words: what is the minimal set of input information that is required to reach the results reported.

4. Could "amons" be a useful chemically motivated set of structures and chemistries that can serve as a common (minimal?) training set that can be enriched with more specific information for a given task?

5. The two sentences "Sampling molecules using conditions that are missing...properties of interest." on p. 5 need clarification. In the first sentence it is not entirely clear what is missing in the training set: is it "molecules" or "conditions" (i.e. properties)? Secondly, depending on the answer to the first question the second sentence appears to contradict the first sentence. If "conditions" are missing in the training data no ML-based method can generate such conditions/properties.

6. Would in true "inverse design" the target property not be part of the loss function? In other words, would one not want to include one or several hyperparameters in the loss function that explicitly bias

the final distribution to the target task? At present this appears to be achieved by restricting the training set to the complement of the target. In other words, one trains on (6<DG<8; with DE as the HOMO-LUMO gap) and observes what structures/chemistries the trained model returns. Some of the resulting structures then "happen" to be outside the training set (which is quite expected) - they are enriched (which is nice - see peaks in Figure 3b) - but is this really "deep"?

In summary, the authors present a method that can learn an enriched distribution with given properties (e.g. HOMO-LUMO gap) by training on the complement and by sampling molecular structures and chemistries. At present it is unclear under what circumstances the present approach is beneficial over straightforward generation and evaluation of a desired chemical composition. At this stage publication is not recommended.

Response to reviewer comments

We thank all reviewers for their thoughtful comments, which helped us to further improve our paper. We have adapted the manuscript according to the raised questions and incorporated the suggestions provided by the expert reviewers. We have updated Figure 1 to include a schematic depiction of the atom placement loop to support the provided math with an easy-to-grasp explanation of the underlying concept. Beyond that, we have evaluated the computational cost of our method compared to exhaustive enumeration of graphs and subsequent DFT relaxation. Here, we show that our generative ML model can significantly reduce the computational cost compared to a naive exploration scheme - even for the small organic molecules considered in this work. Furthermore, we have added the Supplementary Figure S3, where we demonstrate how cG-SchNet generates different conformers for $C_7O_2H_{10}$ isomers.

Changes in our revision have been marked in blue. In the following, we provide a detailed response to the reviewer comments.

Reviewer #1 (Remarks to the Author):

Summary:

The authors propose a conditional generative model for 3d molecular structures cG-SchNet. The model is conditioned on specified structural and chemical properties and the generated samples tend to have such properties in practice. The utility of the method is illustrated in several example problems where new molecules are generated with specific motifs, low energy values, and jointly targeting multiple electronic properties beyond the training regime.

What are the noteworthy results?

The main noteworthy result is the successful extension of a previous unconditional generative model of 3D molecular structures, G-SchNet, to the conditional setting. Previously, targeted generation with G-SchNet required to bias the method by fine-tuning it on a fraction of the training data set with desired properties. This created problems because usually interesting regions for exploration are often those where data is sparse. The new conditional model learns conditional distributions depending on structural or chemical properties and should in principle be less affected by that problem. Also cG-SchNet can jointly target multiple properties without the need to retrain or constrain the sampling process. cG-SchNet makes more efficient use of data resulting in increased generalization.

Will the work be of significance to the field and related fields?

The improvements over G-SchNet are significant and the field will be interested in this work. It seems to me that this is one of the first works to address the problem of conditional generation of 3D structures given properties of interests and, therefore, a significant contribution to the field.

How does it compare to the established literature? If the work is not original, please provide relevant references.

The experimental comparisons are mainly with respect to G-SchNet and the gains seem significant. There are no comparisons to other existing methodology, probably because other approaches do not work by conditioning on properties but instead optimize a reward that could balance different desired properties, including the energy of the resulting molecular configuration. The limitation of the proposed method is that it needs a dataset of 3D structures to be trained, while the reinforcement learning based methods such as [51,52] do not.

Does the work support the conclusions and claims, or is additional evidence needed?

The experiments show that cG-SchNet can sample particularly stable, low-energy $C_7O_2H_{10}$ isomers discovering molecules and motifs that are absent from the QM9 database. cG-SchNet is also used to sample low-energy molecules with small HOMO-LUMO gaps from a domain that is only sparsely represented in the training data. These experiments are enough to show that the proposed method seems to work well in practice.

Are there any flaws in the data analysis, interpretation and conclusions? Do these prohibit publication or require revision?

I did not spot any flaws in the paper. I would encourage the authors to address my comments on the clarity of the molecule generation process as described below.

Thank you for the positive feedback and suggestions. We have updated Fig. 1c to depict the molecule generation process more clearly.

Is the methodology sound? Does the work meet the expected standards in your field?

The methods section describes sound methodology. The authors mention at the end a limitation of the G-SchNet-style approach, which has a cost that is quadratic in the number of atoms. This will prevent the method from working well in large molecules. Another limitation seems the discretization used to sample the atom locations. It is unclear how this can affect performance in practice.

We have added a paragraph where we compare the demand of training, generation, and reference calculations for the experiment with low-energy molecules exhibiting a small HOMO-LUMO gap (page 6-8). We observe that for these small organic molecules, the currently quadratic costs of the generation are negligible in contrast to the computational demand of reference calculations for relaxation. The introduction of a cutoff that limits the number of atoms queried during generation suffices to allow for linear scaling when working with larger molecules.

Concerning the discretization, the chosen resolution of 0.05 Angstrom is sufficient to generate diverse, valid, and target-dependent structures across all experiments. Thus, we conclude that the discretization does not systematically limit the method in practice. The computational complexity is constant as the same grid can be used

across all steps regardless of the number of atoms in the molecule and the generation is relatively fast (e.g. more than 100 low-energy molecules with small HOMO-LUMO gap per second).

Is there enough detail provided in the methods for the work to be reproduced?

I found the description of the sampling process to be confusing. Especially the part on the conditional generation of molecules in appendix E. From the paper, it is hard to understand what is the connection between equation 22 and equation 14. It seems that the authors multiply all the probabilities for each particular grid cell as given by conditioning on all past atoms and then normalize. What is the justification for this?

I think the paper would benefit from having a figure showing a cartoon description of how the molecule generation process works. The figure could include several vignettes corresponding to different steps of the process.

We thank the reviewer for this suggestion. We have assembled such a schematic depiction of the sampling process and added it to Fig. 1. The figure illustrates the molecule generation process and we believe that it is a valuable source of information that makes it easier to convey the general concept of our method.

Regarding the relation between the mentioned equations 14 and 22, the steps 3-5 in the added figure describe how the probabilities are reconstructed from the predictions of pairwise distances that we get from the model. It is correct that the probability in each grid cell is given by multiplication of the corresponding pairwise distance probabilities to all past atoms. We introduced this approximation with G-SchNet as it allows us to construct a rotationally equivariant positional distribution while respecting local and global invariances of molecular structures (e.g. rotation of groups, indexing of atoms, etc.). We have reported how the approximation can suffer from artifacts in highly symmetric sub-structures and how this is alleviated by the use of the auxiliary focus token, which breaks such symmetries (see [48]). Overall, the distribution is computationally cheap to obtain and has shown to reasonably capture the characteristics of molecular structures in our experiments. Thus, it is suitable for targeted discovery of novel molecules in practice. We have added this information to the manuscript.

Some minor comments:

The authors should verify that references to arxiv papers cite instead the actual published source.

Figure 1 b -> c

isomers exhibits HOMO-LUMO -> isomers exhibit HOMO-LUMO

We have corrected these mistakes and updated the references.

Reviewer #2 (Remarks to the Author):

The manuscript presents a generative neural network architecture for 3D molecular structures that can be biased towards specific chemical properties and molecular motifs. The authors demonstrate that the model can generate new organic molecules biased towards specific atomization energies, motifs, and other chemical properties.

This manuscript is well written and would be valuable to the scientific community as is. However, additional experiments involving the interactions highlighted by the authors in the introduction to be challenging to solve without a coordinate-based model (long-range interactions, conformers, metallic complexes, etc.) would significantly strengthen this work. The current experiments only involve relatively simple organic chemistry and can be done with a graph or smiles-based generative model. Aside from this concern, I would recommend this work for publication in Nature Communications.

Inspired by your suggestions, we included a brief analysis of conformers found for the five $C_7O_2H_{10}$ isomers most often generated by our model (Fig S3 in the Supplements). It sketches a path for conformer search with cG-SchNet as an example application where coordinate-based models are required. We expect that extensive experiments with systems which require coordinate-based models will be a fruitful direction for future work on 3d molecule generation evolving from our presented results.

Reviewer #3 (Remarks to the Author):

Review of Gebauer et al.

This manuscript presents a machine learning-based approach to generate molecular structures with given stoichiometry that exhibit particular properties. The ansatz pursued is that of a conditional generative neural network.

We want to stress that the generation of molecules with given stoichiometry is only one possible application of our conditional model. However, the atomic composition does not need to be part of the target properties (see e.g. the experiments with isotropic polarizability in Fig. 1b, with fingerprints in Fig. 3a, or with low-energy + small HOMO-LUMO gap in Fig. 5). We have clarified this in the manuscript, e.g. by adding the following paragraph:

“Our architecture is designed to generate molecules of arbitrary size and does not require the specification of a target composition. Consequently, it learns the relationship between the composition of molecules and their physical properties in

order to sample the most suitable candidates for a given target property, e.g. preferring smaller structures when targeting small polarizabilities.”

Detailed comments:

1. It appears that the ansatz pursued assumes that the emergence of a chemical property, e.g. the HOMO-LUMO gap, is a smooth function of chemical composition and molecular structure. This, however, is not necessarily the case. It would be of interest to report the HOMO-LUMO gap along such a "chemical trajectory" as indicated on by the construction procedure in eq. (2) and see how the target property "develops" or "emerges" as the molecule grows. Or does the model just need to "cover" the desired range of observable with given chemistry?

Our method does not require intermediate evaluations of a target property for partial structures. Instead only the expected final property is input to the generation procedure. Accordingly, only the property of interest of a full molecule is required during training. Therefore, our model does not have to rely on smoothness assumptions of the target property w.r.t. a “chemical trajectory”.

We have added Fig. 1c to clarify the generation process.

2. P. 4 states that "...Thus, the model can only learn to generate molecules with gaps outside this range from other compositions." What are these "other compositions" in the present case?

This statement relates to the design of the experiment regarding the generalization capabilities of our model. In the corresponding experiment, the training data consists of only a few $C_7 N_1 O_1 H_{11}$ molecules with HOMO-LUMO gap values between 6 eV and 8 eV and many structures of other compositions that are randomly sampled from QM9 and cover the whole range of gaps in the data set (i.e. also smaller than 6 eV and larger than 8 eV). I.e., the model has to transfer the knowledge about structures with gaps smaller than 6 eV or larger than 8 eV from other compositions than $C_7 N_1 O_1 H_{11}$ to the target composition when we provide gap values of 5 eV and 9 eV and the composition $C_7 N_1 O_1 H_{11}$ as conditions for sampling .

We designed the task in this way as it is a particularly challenging generalization for neural networks: one might expect that the model predominantly samples either $C_7 N_1 O_1 H_{11}$ molecules with intermediate gap values or fails to sample the target composition when building structures with more extreme gap values, as this is what it has observed in the training data. However, the results show that the model is successfully able to generalize to this new set of target property values that was never observed in the training data as it predominantly samples $C_7 N_1 O_1 H_{11}$ molecules close to the extreme gap targets. We have updated this part of the paper to increase its clarity:

“We restrict the training data consisting of 55k molecules from QM9 to contain no $C_7 N_1 O_1 H_{11}$ isomers with HOMO-LUMO gap values outside the intermediate

range (Fig. 3b). Thus, the model can only learn to generate molecules with gaps outside this range from compositions other than $C_7 N_1 O_1 H_{11}$.

3. The results suggest that the model increases the probability to generate molecular structures and chemical compositions that contain the desired target property. The question is whether the procedure proposed speeds up discovery relative to exhaustive sampling of all graphs. Given that DFT calculations are computationally inexpensive it may well be that generating the ML-based model, training it, evaluating and testing it is considerably more expensive than straightforward generation of the necessary graphs and evaluating and filtering for the desired property. A careful discussion of this question is required and where the "break-even point" for given cost/benefit is. In other words: what is the minimal set of input information that is required to reach the results reported.

We agree that this is an important consideration which we missed to report on before. We have now carefully measured the time for training, generation, and DFT reference calculations for our experiment with low-energy molecules with small HOMO-LUMO gap. Although DFT calculations are comparably fast for the small organic molecules in QM9, the reference calculations still remain the clear bottleneck in the process. Thus, the question is how many reference calculations are needed to obtain and identify the desirable molecules. We found that in the considered case where we target low-energy molecules with small gap values, i.e. property values at the border of the QM9 distribution, the model already significantly increases the efficiency even for these small structures. We report:

"The efficiency of cG-SchNet in finding molecular structures close to the target conditions is particularly evident compared to exhaustive enumeration of graphs with subsequent relaxation using DFT. In both cases, the relaxation required to obtain equilibrium coordinates and the physical properties is the computational bottleneck and takes more than 15 minutes per structure for the molecules generated in this experiment. Furthermore, the calculation of the internal energy at zero Kelvin (U_0) requires additional 40 minutes per molecule. In contrast, the generation with cG-SchNet takes only 9 milliseconds per structure on a Nvidia A100 GPU when sampling in batches of 1250. The training time of about 40 hours is negligible, as it corresponds to the relaxation and calculation of U_0 of only 44 structures. Thus, the efficiency is determined by the number of molecules that need to be relaxed for each method. The QM9 data set was assembled by relaxing structures from the GDB enumeration [61] of graphs for small organic compounds. Of the ~78k molecules that we did not use for training, 354 molecules are close to the target region. Relaxing only the 5283 structures proposed by cG-SchNet, i.e. less than 10% of the computations performed by screening all graphs, we can already recover 46% of these structures. Additionally, the model has unveiled valid molecules close to the target that are not contained in the data set. More than 380 of these are larger than QM9 structures and thus not covered. However, 253 smaller structures were missed by the enumeration method."

This shows that an exhaustive exploration by sampling graphs is non-trivial, even for such small structures. Going to larger molecules will further aggravate this problem, as the number of possible graphs explodes. Therefore, a guided exploration as with our *cG-SchNet*, which effectively limits the number of candidates, becomes the only feasible option. We updated our discussion accordingly:

“Furthermore, we have sampled more than 800 low-energy molecules with HOMO-LUMO gaps smaller than 4.5 eV from a domain that is only sparsely represented in the training data. Although the exploration of such small molecules with exhaustive sampling of molecular graphs and subsequent evaluation with density functional theory is computationally feasible, our model considerably accelerates the process by providing reasonable candidate structures.”

4. Could "amons" be a useful chemically motivated set of structures and chemistries that can serve as a common (minimal?) training set that can be enriched with more specific information for a given task?

Indeed, “amons” or other fragmentation approaches can provide initial training data, in particular when targeting larger structures. We have updated our discussion to include this idea:

“While the small organic compounds considered in this work are well represented by QM9, the model might benefit from enhancing the training data using representative building blocks such as ‘amons’ [63] or other fragmentation methods [64, 65]. This becomes increasingly important when tackling larger molecules where reference data is hard to obtain.”

5. The two sentences "Sampling molecules using conditions that are missing...properties of interest." on p. 5 need clarification. In the first sentence it is not entirely clear what is missing in the training set: is it "molecules" or "conditions" (i.e. properties)? Secondly, depending on the answer to the first question the second sentence appears to contradict the first sentence. If "conditions" are missing in the training data no ML-based method can generate such conditions/properties.

Here we refer to molecules with specific target property values that are missing in the training data (i.e. a very small HOMO-LUMO gap and low energy). We have clarified this part:

“The ability to sample molecules that exhibit property values which are missing in the training data is a prerequisite for the targeted exploration of chemical space. A generative model needs to fill the sparsely sampled regions of the space, effectively enhancing the available data with novel structures that show property values of interest.”

6. Would in true "inverse design" the target property not be part of the loss function? In other words, would one not want to include one or several hyperparameters in the loss function that explicitly bias the final distribution to the target task? At present this appears to be achieved by restricting the training set to the complement of the

target. In other words, one trains on ($6 < DG < 8$; with DE as the HOMO-LUMO gap) and observes what structures/chemistries the trained model returns. Some of the resulting structures then "happen" to be outside the training set (which is quite expected) - they are enriched (which is nice - see peaks in Figure 3b) - but is this really "deep"?

The target property is part of the loss function implicitly, since it is given to the model as input, which is the crucial benefit of our conditional approach. The model learns the conditional distribution of molecules in a general training phase, based on both the structures and their properties. Afterwards, one can specify any combination of conditions (i.e. values of the target properties) to sample molecules that are highly likely to exhibit the desired conditions.

Previous approaches for 3d molecules, which do not handle conditions, need to be trained for one specific set of target properties by carefully adjusting the training data (e.g. biased G-SchNet) or the loss function/objective (e.g. the reinforcement learning-based methods). Those models *cannot* be used to target other property values without retraining them with adjusted data or loss. Since this is an important advantage of our approach, we have adapted a few sentences in the introduction to further stress and clarify the difference between conditioning (our method) and biasing/training (previous methods):

“Previously proposed methods have been biased towards one particular set of target property values at a time by adjusting the training objective or data [48, 51]. In contrast, our conditional approach permits searching for molecules with any desirable set of target property values after training is completed. It is able to jointly target multiple properties without the need to retrain or otherwise indirectly constrain the sampling process. This provides the foundation for the model to leverage the full information of the training data resulting in increased generalization and data efficiency.”

Furthermore, there is no need to train the model on the “complement” of the target, but targeting molecules that have property values which were left out during training is a comparatively hard generalization task. We designed some of our experiments to target the “complement” of the training data only to demonstrate the generalization capabilities of cg-SchNet. The model is able to sample structures exhibiting the specified target values, exactly because it captures the conditional distribution - i.e. the differing probability of certain molecules and motifs depending on the provided target property values.

In summary, the authors present a method that can learn an enriched distribution with given properties (e.g. HOMO-LUMO gap) by training on the complement and by sampling molecular structures and chemistries. At present it is unclear under what circumstances the present approach is beneficial over straightforward generation and evaluation of a desired chemical composition. At this stage publication is not recommended.

We have adapted the manuscript in order to increase its clarity and its advantages over unconditioned methods. Furthermore, we discuss additional results that

compare the computational complexity of our method to exhaustive enumeration and relaxation of graphs, where we see that the proposed generative model is already more efficient than the exhaustive approach. We are happy that the manuscript has significantly improved thanks to the excellent and helpful suggestions by the reviewers.

REVIEWER COMMENTS

Reviewer #3 (Remarks to the Author):

Re-review of Gebauer et al.

The authors have addressed and clarified some of the points raised by the referees. One open point is still in what sense the present work is related to "inverse design" and how significant this progress is.

1. A concise statement of the problem being solved is required. Is it to find "the best molecule with a particular property" or is it to suggest "families or classes of molecules with a specific property"?

Or something else?

2. The authors stress that multi-property optimization is possible with the present approach. Here, the two properties considered are the HOMO-LUMO gap and the relative atomic energy. In Figure 4d: the authors explain that the energy values indicate "...comparatively low energy..". As the molecules all have identical chemical composition it would be more transparent to report the energy difference with respect to either a) the lowest energy structure encountered or b) the largest stabilization energy. What does a "target energy of -0.1 eV" mentioned in the caption actually mean?

3. The caption for Figure 4d states that "Relaxed example low-energy isomers generated by cG-SchNet.." are reported. How do these

structures change when re-optimized with quantum chemistry? By how much do structures and energies change? In other words, how "good" are the relaxed structures predicted by cG-SchNet when compared with results from electronic structure calculations?

4. How does the approach extend when additional properties (polarizabilities, dipole moment, etc.) included? While one can expect that cG-SchNet is extensible in these regards, the question is of what value this will be in practice because molecules with optimal "property 1" will potentially be "in the middle of the distribution for property 2". In other words: finding a molecule for which all properties are "optimal" is probably neither realistic nor meaningful. Hence the request to define the task - see point 1 above.

5. With respect to point 1 above: what is the molecule with the largest / smallest HOMO-LUMO gap predicted by the present model and how does this compare with the results from electronic structure calculations? Of course, this quest is only meaningful if the molecules with the largest/smallest HOMO-LUMO gaps are not included in the test set. This question concerns the extrapolation behaviour of the model.

6. The authors write that the model needs "..to fill the sparsely sampled regions.." which targets the interpolation properties of the model. Is it by now not accepted that this "interpolation" in general is possible, e.g. from the work by von Lilienfeld? What appears more challenging is extrapolation. Discussing this point is required.

7. One possibility for an overarching test would be to plot the HOMO-LUMO gap vs. difference in total energy between the molecules and the lowest energy structure for a) the model and b) for electronic structure calculations. This could be separated into training set and randomly drawn samples. Such a plot would cover both, interpolation and extrapolation of the model.

While there is no doubt that the work presented is of high quality the question still remains how significant the step forward really is. Much of this depends on what precisely the quest pursued is and how the model performs away "from its comfort zone", i.e. for extrapolation.

Response to reviewer comments

We thank the reviewer for the additional comments and questions. Changes in our revision have been marked in blue. In the following, we provide a detailed response to the reviewer comments.

Reviewer #3 (Remarks to the Author):

Re-review of Gebauer et al.

The authors have addressed and clarified some of the points raised by the referees. One open point is still in what sense the present work is related to "inverse design" and how significant this progress is.

1. A concise statement of the problem being solved is required. Is it to find "the best molecule with a particular property" or is it to suggest "families or classes of molecules with a specific property"? Or something else?

The goal of our work is to find a set of candidate molecules with specific properties. As stated in the abstract, the model "enables targeted sampling of novel molecules from conditional distributions, even in domains where reference calculations are sparse." This is achieved by learning a distribution of molecular structures, where molecules with the desired properties have a higher probability of being drawn. For an excellent and extensive introduction to various approaches of molecular inverse design, including deep generative models, we refer to Sanchez-Lengeling et al. (Science 361, 360–365, 2018) in the paper. We have further clarified the motivation of our approach in the introduction as follows:

"Generative ML models have recently gained traction as a powerful, data-driven approach to inverse design as they enable sampling from a learned distribution of molecular configurations [29]. By appropriately restricting the distributions, they allow to obtain sets of candidate structures with desirable characteristics for further evaluation. [...] Consequently, [our model] learns the relationship between the composition of molecules and their physical properties in order to sample candidates exhibiting given target properties, e.g. preferring smaller structures when targeting small polarizabilities."

2. The authors stress that multi-property optimization is possible with the present approach. Here, the two properties considered are the HOMO-LUMO gap and the relative atomic energy. In Figure 4d: the authors explain that the energy values indicate "...comparatively low energy..". As the molecules all have identical chemical composition it would be more transparent to report the energy difference with respect to either a) the lowest energy structure encountered or b) the largest stabilization energy. What does a "target energy of -0.1 eV" mentioned in the caption actually mean?

We have defined the relative atomic energy as the total energy minus the average energy of molecules of the same composition and normalized by the number of atoms. Similar to the atomization energy, this removes the single-atom energy

contributions and allows us to treat molecules of different size and composition in a comparable and normalized manner. However, instead of the single atom energy in vacuum, we use the mean energy of an atom in molecular environments according to the reference data. Therefore, the target of -0.1 eV means that the conformations sampled shall have 0.1 eV lower energy per atom than the average arrangements of atoms with that composition in QM9. We explain the relative atomic energy in detail in Supplementary Methods X B. Note that Zubtyuk et al. (*Science Advances* 5(8): eaav6490, 2019) have defined a similar energy for their neural network potential AIMNet.

The distribution of the relative atomic energy of C7O2H10 structures in QM9 is plotted in Figure 4a (orange line) and puts the target energy of -0.1 eV into context: it is a low value at the tail of the distribution, i.e. close to the C7O2H10 isomers with the lowest energy in QM9. Although it would be possible to change the reference point in the evaluation as suggested, we are convinced that it is most transparent to report the values of the property that the model is conditioned on, i.e. the relative atomic energy. Otherwise, the relationship between the energy used during training and the energy depicted in the plots might confuse the readers. We added further explanation to better clarify our choice and to avoid potential misunderstanding.

3. The caption for Figure 4d states that "Relaxed example low-energy isomers generated by cG-SchNet.." are reported. How do these structures change when re-optimized with quantum chemistry? By how much do structures and energies change? In other words, how "good" are the relaxed structures predicted by cG-SchNet when compared with results from electronic structure calculations?

The changes of structures before and after relaxation have been reported for all our experiments in Supplementary Table 2. We measure the spatial change between conformations before and after re-optimization (i.e. relaxation) with the root-mean-square deviation (RMSD) of the atom positions. For the molecules from the experiment depicted in Fig. 4d, the median of the RMSD is 0.26 Angstrom. In order to further visualize these results, we have added Supplementary Figure 4, where we plot generated C7O2H10 conformations versus the closest equilibrium found by relaxation and report the corresponding RMSD per structure. Here we see that the generated structures are close to results from electronic structure calculations and the deviations can mostly be attributed to rearrangements of hydrogen atoms.

4. How does the approach extend when additional properties (polarizabilities, dipole moment, etc.) included? While one can expect that cG-SchNet is extensible in these regards, the question is of what value this will be in practice because molecules with optimal "property 1" will potentially be "in the middle of the distribution for property 2". In other words: finding a molecule for which all properties are "optimal" is probably neither realistic nor meaningful. Hence the request to define the task - see point 1 above.

As demonstrated in Fig. 5, the cG-SchNet shifts the unconditional distribution of molecules such that the specified combination of properties becomes more likely during sampling. Of course, if the chosen target is impossible to realize, the model will yield structures with various trade-offs. Note that it is a known limitation of conditional generative models for molecular inverse design that these trade-offs can

not be specified (see for example the graph and SMILES based models by Li et al. (*Journal of Cheminformatics* 10(1): 1-24, 2018), Lim et al. (*Journal of Cheminformatics* 10(1): 1-9, 2018), or Polykovskiy et al. (*Molecular Pharmaceutics* 15(10): 4398-4405, 2018)). An extension to overcome these shortcomings, e.g. by explicitly sampling along a Pareto front, is an interesting direction for future work. In practice, we combine cG-SchNet with a fast neural network potential in order to filter the generated candidates according to the desired target property trade-off. We updated the discussion to further clarify this point.

5. With respect to point 1 above: what is the molecule with the largest / smallest HOMO-LUMO gap predicted by the present model and how does this compare with the results from electronic structure calculations? Of course, this quest is only meaningful if the molecules with the largest/smallest HOMO-LUMO gaps are not included in the test set. This question concerns the extrapolation behaviour of the model.

In our experiments, the model is not conditioned to find “the one molecule with the smallest HOMO-LUMO gap” but rather “a set of suitable candidates with HOMO-LUMO gap close to the target value” (i.e. small relative to the training examples). In this light, it appears more relevant to check how many unique and valid 3d structures close to the target property value(s) are found. We report the structures sampled closest to the targets for all experiments (see e.g. Figures 4 and 5). Beyond that, we have added Supplementary Figure 5, which shows examples of molecules with particularly low HOMO-LUMO gap for both the model and the training set (see our response to point 7 below for more details).

6. The authors write that the model needs “..to fill the sparsely sampled regions..” which targets the interpolation properties of the model. Is it by now not accepted that this “interpolation” in general is possible, e.g. from the work by von Lilienfeld? What appears more challenging is extrapolation. Discussing this point is required.

Whether one is in the extrapolation or interpolation regime depends on the feature space and which assumptions are made about its structure. cG-SchNet can efficiently exploit the structure of the feature space by leveraging rotational and translational symmetries as well as locality assumptions. Although inference from small to large molecules is often considered extrapolation, it may actually correspond to interpolation in the cG-SchNet feature space. Therefore, we rather prefer the term “generalization” over “extrapolation” in this context.

Note that we have focused on this very aspect in the presented experiments. We extensively show the generalization capabilities of our model by using conditions that were either missing in the training data (e.g. a specific composition, a combination of composition and HOMO-LUMO gap value, a novel fingerprint) or are at the borders of the training distribution (e.g. small HOMO-LUMO gap and energy or extreme polarizability values) and observe that novel structures with traits that are missing within QM9 are sampled (e.g. C7O2H10 with carboxylic acid groups or molecules with more than 9 heavy atoms for large polarizability). We have clarified this in the discussion.

7. One possibility for an overarching test would be to plot the HOMO-LUMO gap vs. difference in total energy between the molecules and the lowest energy structure for a) the model and b) for electronic structure calculations. This could be separated into training set and randomly drawn samples. Such a plot would cover both, interpolation and extrapolation of the model.

This kind of evaluation is indeed illuminating and can be found in Figure 5a. Note, however, that we use the relative atomic energy, as described above, which has the same information content. The HOMO-LUMO gap vs the relative atomic energy are plotted for both the training data (QM9, left-hand side) and molecules generated with our model (right-hand side). Our evaluation shows that the model generalizes well as it unveils more structures in the target region at the border of the QM9 distribution than can be found in QM9 itself. The goal of the experiment depicted in Figure 5 is to find molecules with low HOMO-LUMO gaps (relative to the training data) while avoiding undesirable, unstable motifs (which we observed when sampling low HOMO-LUMO gap structures with the previous G-SchNet model). We achieve this by conditioning on the relative atomic energy, which relates the energy of molecules to other isomers of the same composition. This energy is usually higher, when such unstable motifs are present. We added further clarification to Figure 5 and the corresponding subsection in the results.

Furthermore, we have added Supplementary Figure 5, where we zoom in on the target region and show individual points (i.e. the gap and energy of individual molecules) instead of plotting the density. Here one can assess the interpolation and extrapolation in detail: while the conditioning target has been chosen at the edge of the training distribution, the neural network extrapolates to candidate molecules with lower relative atomic energy *and* HOMO-LUMO gap.

While there is no doubt that the work presented is of high quality the question still remains how significant the step forward really is. Much of this depends on what precisely the quest pursued is and how the model performs away "from its comfort zone", i.e. for extrapolation.

As acknowledged by Reviewer #1, cG-SchNet is "one of the first works to address the problem of conditional generation of 3D structures given properties of interests". The generalization performance of our model is thoroughly evaluated and discussed (see answer to point 6 above). All experiments involve conditions which are rare or even unseen by the model during training, i.e. "away from its comfort zone". On this basis, we find that cG-SchNet is ready to be employed in practice for generating candidate molecular structures with desired properties that generalize beyond the reference data. We therefore firmly believe that cG-SchNet is indeed a highly significant step forward in the inverse design of 3D molecules with generative ML models.

REVIEWERS' COMMENTS

Reviewer #3 (Remarks to the Author):

Re-review of Gebauer et al.

The authors have addressed some of the additional points raised. Nevertheless, open questions remain.

1. The value of 0.1 for the "target energy" appears to be universal across all elements and for each atomic species (H, C, N, O,..) in all chemical environments encountered. This is unrealistic, see e.g. Unke and Meuwly, JCP 2018, who also analyzed QM9 in this regard.
2. In the reply to query 3 of reviewer 3 the authors report the structural RMSD. But how do the minimum energies and the HOMO-LUMO gaps change/differ between the relaxed structure predicted by cG-SchNet and by quantum chemistry?
3. While Figure S5 provides an impression of the generalization capabilities, the 4 molecules contain 10 instead of 9 heavy atoms and thus provide only limited information on the broader question of generalization.
4. Although touched upon on p. 8, it is not really clear that the present approach is truly more effective in terms of a) overall computer time for the entire workflow and b) improved accuracy over a more standard approach to determine candidate molecules falling into specific target ranges for certain observables.

Response to reviewer comments

We thank the reviewer for the additional comments. Changes in our revision have been marked in blue. In the following, we provide a detailed response to the reviewer comments.

Reviewer #3 (Remarks to the Author):

Re-review of Gebauer et al.

The authors have addressed some of the additional points raised. Nevertheless, open questions remain.

1. The value of 0.1 for the "target energy" appears to be universal across all elements and for each atomic species (H, C, N, O,...) in all chemical environments encountered. This is unrealistic, see e.g. Unke and Meuwly, JCP 2018, who also analyzed QM9 in this regard.

In the cited work, the contributions of atomic species and chemical environments to the energy is analyzed in absolute terms. This is different from our approach, where we use a relative atomic energy as explained in our previous response and in Supplementary Methods 2:

"We define a relative atomic energy that describes whether the energy per atom of a 3d conformation is comparatively high or low with respect to other structures in the data set that share the same atomic composition [...]. In this way, we can [...] treat molecules of different size and composition in a comparable and normalized manner. This allows our model to learn a relation between 3d conformations and their energy that can be transferred across compositions, as can be seen in our experiments where we sample low-energy C₇₀H₁₀ isomers with a model that was trained solely on other compositions (see Figure 4 in the paper)."

The reference point for the relative atomic energy depends on the atomic composition which makes the energy of molecules with different compositions comparable. The range of the relative atomic energy varies for different compositions and therefore the target value of -0.1 eV is not universal but specifically chosen for C₇₀H₁₀ isomers by checking the range in QM9 (see Figure 4a). In the experiments with low-energy structures with small HOMO-LUMO gap, we instead use a target energy of -0.2 eV as the composition is not fixed in this setting and therefore the range is increased.

2. In the reply to query 3 of reviewer 3 the authors report the structural RMSD. But how do the minimum energies and the HOMO-LUMO gaps change/differ between the relaxed structure predicted by cG-SchNet and by quantum chemistry?

The goal of cG-SchNet is to obtain a limited set of candidate structures that are highly likely to exhibit the target properties in order to reduce the amount of quantum chemistry reference calculations when exploring chemical space. We assume that in practical settings the structures obtained with our model are subsequently relaxed with DFT or neural network potentials for further evaluation and selection. Therefore, we report the energy and HOMO-LUMO gap of relaxed structures. The precise energy and HOMO-LUMO gap of raw, sampled structures is not relevant as long as the molecules do not drastically change during relaxation. To verify this, we report the structural RMSD, which shows that “cG-SchNet samples molecules which are close to equilibrium configurations and thus require only a few steps of relaxation with DFT or a neural network potential”. We have clarified this in the manuscript.

3. While Figure S5 provides an impression of the generalization capabilities, the 4 molecules contain 10 instead of 9 heavy atoms and thus provide only limited information on the broader question of generalization.

We did not consider the number of atoms when picking the example structures in Supplementary Figure 5. However, cG-SchNet also samples novel molecules with 9 heavy atoms and with gap and energy outside the training data distribution. We have added such a structure to the figure.

Note that the step from 9 to 10 heavy atoms and how it exactly influences energy and HOMO-LUMO gap is not explicitly represented in the training data. Therefore, sampling molecules with more than 9 heavy atoms that meet the target properties is a prime example of generalization. We have clarified this in the manuscript.

4. Although touched upon on p. 8, it is not really clear that the present approach is truly more effective in terms of a) overall computer time for the entire workflow and b) improved accuracy over a more standard approach to determine candidate molecules falling into specific target ranges for certain observables.

Our evaluation contains a comparison of finding low-energy molecules with small HOMO-LUMO gap with cG-SchNet and with exhaustive enumeration of graphs including subsequent DFT relaxation. We report the average computation time of training cG-SchNet (40 hours), sampling a molecule with cG-SchNet (9 ms), relaxing a molecule with DFT (15 min), and calculating U0 of molecules with DFT (40 min) for that experiment. As the relaxation of molecules is clearly the bottleneck, we compare the number of structures that need to be relaxed and how many target molecules can be obtained in both approaches. We find that “we obtain more than two times the amount of molecules close to the target property values with cG-SchNet than with the exhaustive enumeration method while requiring less than 10% of the computation time.” We have clarified this in the manuscript to pronounce that cG-SchNet is, in this instance, clearly more effective in terms of overall computing time and the amount of found structures that exhibit target properties. Beyond that, a comparison of the effectiveness of cG-SchNet to different established methods in various fields, including different properties and varying target ranges, is an important direction for future work and we have added this point to the discussion.